# Reinforcement Learning for Symbolic Equation Solving

## Abstract

We present a reinforcement-learning agent that solves symbolic equations step by step —
both nonlinear *closed* equations (radicals, exponentials, trigonometric) and, for the first
time, *restricted-open* families requiring a change of variables (CoV) such as completing the
square. We cast algebra as an MDP with a dynamic action space and a tree-structured policy
(TreeMLP) that learns the full solution procedure from reward alone, with no supervised
solution traces; the CoV substitution comes from a supervised generator interchangeable
with a CAS call ($\leq 0.02$ end-task effect), and success means one root verified by substitution
and mapped back by exact inverse composition. On closed equations the agent matches the
prior best on CommonCore (0.93 greedy vs. ConPoLe's 0.925) and handles nonlinear families
(radical, log, trig) under a single policy. On restricted-open quadratic, cubic, quartic, and
exponential families it reaches 0.79 beam / 0.67 greedy (5-seed means); the greedy mean
alone exceeds the strongest non-learned search ($A^*$, 0.64), with beam adding a further
margin. We test the CoV trigger directly against hand-coded rules: on the three single-CoV
families a one-line detector matches or beats the policy — nothing is learned there — but
on the exponential family, which requires a *nested* CoV, the natural rule solves none of
the held-out equations while the policy solves 75% from reward alone. The honest claim is
*recovery* of a hand-engineered trigger without access to the detector, not superiority over
it. At $10\times$ scale a sharp seed-level bimodality emerges; a UCB learning-progress curriculum
shows a non-significant positive trend toward mitigating it.

## 1 Introduction

Most machine-learning approaches to symbolic mathematics map a problem directly to its answer Lample &
Charton (2020): a model emits a solution in one shot. Many downstream goals, however, need the *procedure*:
a sequence of explicit, replayable transformations that reaches the solution and exposes the trace and the
policy that produced it. Differential-equation reductions, theorem-prover tactics, and didactic step-by-step
solvers all consume such traces. We show that reinforcement learning can acquire this procedural capability
for algebraic equation solving: building on RL for nonlinear *closed* equations (O'Keeffe, 2025) (radical,
exponential, trigonometric), our agent additionally solves, to our knowledge for the first time, the harder
*restricted-open* setting, where the solution requires a change of variables (CoV) drawn from hand-designed
families with known closed forms. The main policy learns from reward alone, with no supervised solution
traces; the one supervised component is the CoV substitution generator, which is interchangeable with a
CAS call. Throughout, *solving* means producing one root verified by substitution in the active equation and
mapped back by exact inverse composition.

We formulate symbolic algebra as an MDP whose actions pair an algebraic operation with a sub-expression
of the current equation, the legal-action set re-derived from the state each step. The policy/value network is
TreeMLP: node IDs are embedded, $K$ rounds of strictly-local sum-aggregation message passing update node
states, and a masked pool yields a fixed-size graph feature for the PPO heads. Three light inductive biases
(curriculum, success-replay buffer, constant-relabel macroaction) are layered on top; on CommonCore this
matches the prior best (ConPoLe; Poesia et al. 2021).

We expose CoV as a single macroaction in the same dynamic action space; the main policy learns the
surrounding procedure and selects the CoV trigger from reward. We test that trigger directly against hand-

coded rules rather than asserting it (Section 4.2.3): on the three single-CoV families a one-line detector matches or beats the policy — nothing is learned there. The trigger has content only on the exponential family, which requires a *nested* CoV, where the natural rule solves 0/75 and the policy solves 56/75 from reward alone. The honest claim is *recovery* of a hand-engineered trigger without access to the detector, not superiority over it. At $10\times$ scale a sharp seed-level bimodality emerges; a UCB learning-progress curriculum shows a non-significant positive trend toward mitigating it.

**Contributions.** (a) *The first reward-driven RL agent for restricted-open (CoV) symbolic equation solving*, with the trigger recovered from reward alone and composable with closed-equation steps under a single policy. The claim is scoped to *recovery*: on single-CoV families a rule suffices; on the nested-CoV exponential family the policy recovers the trigger without being given the detector. (b) *TreeMLP and supporting machinery*: a tree-structured policy/value backbone together with relabel-constants and success replay — the configuration PPO-TREE-RC-BUF, which matches the prior best on closed CommonCore equations (0.93 greedy vs. ConPoLe's 0.925; Table 1). (c) *A seed-level failure mode and a candidate mitigation*: a sharp bimodality at $10\times$ scale in which a run either learns the task or collapses into a policy trap, with a UCB learning-progress curriculum as a non-significant positive mitigation trend.

## 2 Related work

**RL for symbolic equation solving.** Poesia et al. (2021) introduce ConPoLe for linear equations under low-level axioms (92.5% on CommonCore, our benchmark). Dabelow & Ueda (2024) train an RL agent on a stack-calculator interface for linear manipulation. Most closely related, O'Keeffe (2025) extend RL to nonlinear *closed* equations (radicals, exponentials, trigonometric) with PPO over a graph-structured action space and curiosity-driven exploration. Our work differs in *scope* (we add the open/CoV setting, which requires introducing structure absent from the original equation), *exploration mechanism* (curiosity bonuses fail to match a no-curiosity baseline on top of TreeMLP; we use success replay and constant relabeling instead), and *method* (TreeMLP backbone, value-guided beam, bimodality analysis). The open/CoV capability is what we claim as first; nonlinear-closed RL is due to O'Keeffe (2025).

**Neural symbolic math.** Lample & Charton (2020) train a seq2seq Transformer on $\sim 10^8$ supervised pairs to produce one-shot solutions for integration and ODEs. We learn from a per-step reward with no supervised solution traces; the only supervised component is $\pi_{\text{cov}}$, trained on $\sim 10^3$ known substitutions.

**Symbolic regression and program synthesis.** DSO Petersen et al. (2021) and DreamCoder Ellis et al. (2021) *generate* formulae; we manipulate a given equation by legal transformations. Li et al. (2022) learn high-level actions for equation solving via expert iteration; our CoV macroaction is similar in spirit but is a single concrete substitution rather than an induced abstraction library.

**Graph encoders for symbolic state.** GCN Kipf & Welling (2017) and GraphSAGE Hamilton et al. (2017) assume large undirected neighbourhoods; symbolic expression trees are sparse rooted DAGs with a small ordered vocabulary. TreeMLP drops degree normalisation and learned edge transforms in favour of a strictly local sum-aggregation update aligned with tree semantics (Section 3.1).

**Formal mathematical reasoning.** Peano Poesia & Goodman (2023) and AlphaZero-style theorem provers target proof-style validity rather than computational equation solving; these settings are complementary.

**Curriculum learning and capability collapse.** Matiisen et al. (2019) introduce learning-progress (LP) curricula; our UCB-LP variant adds a UCB1 bonus on per-class sampling weights Auer et al. (2002), structurally similar to ACTOR-CURATOR Gu et al. (2026). The seed-level bimodality we observe is the symbolic-RL instance of bimodal capability acquisition documented in Zhao et al. (2025); Suau et al. (2024); Dong et al. (2025).

## 3 Problem Formulation

Our Markov Decision Process (MDP) formulation is

**States**: equations like $ax + b = 0$ or $cx + d = -x/b$. We vectorize these using preorder traversal over the equation's expression tree (Appendix A.13, Figure 7) and pad to a maximum length $L = 50$. Operations and symbols are mapped to integers, e.g. {add:1, sub:2, mul:3, ..., $x$:5, $a$:6, $b$:7, ...}; both lhs and rhs are encoded and concatenated. For example, $x + a \rightarrow [\text{add}, x, a] \rightarrow [1, 5, 6, \text{PAD}, \ldots]$.

**Actions**: represented as $(operation, term)$ pairs, such as $(\text{sub}, b)$ or $(\text{div}, a)$. Formally,

$$
\begin{aligned}
O = &\{\text{add, sub, mul, div}\} \\
&\cup \{\text{square, sqrt, exp, log, sin, cos, asin, acos}\}, && (1) \\
T = &\text{SubExpr(lhs)} \cup \text{SubExpr(rhs)}, && (2) \\
A = &(O \times T) \\
&\cup \{(\text{expand}, \text{None}), (\text{collect}, x), (\text{mul}, -1)\}. && (3)
\end{aligned}
$$

For the terms, we choose the list of sub-expressions in the lhs and rhs expression trees. Example of sub-expressions are

$$
\begin{aligned}
ax + b = 0 &\Rightarrow \{a, x, ax, b\} && (4) \\
\tfrac{ax+b}{cx+d} + e = 0 &\Rightarrow \{a, x, ax, b, c, d, cx, cx + d, ax + b\} && (5)
\end{aligned}
$$

This term set is expressive enough to solve rational equations and all other equation types we consider in this paper. It is also dynamic: the list of sub-expressions is derived from the equation/state and thus has variable length. We also include $(\text{mul}, -1)$ and

$$
\begin{aligned}
\text{expand:} \quad & cx + d + e(ax + b) \rightarrow aex + be + cx + d \\
\text{collect } x\text{:} \quad & ax + bx \rightarrow (a + b)x
\end{aligned}
$$

These allow the agent to perform key algebraic steps (Appendix A). Finally, we index $A$ serially and cap it at $|A| = 50$ (the largest set any dataset equation produces is $\sim 40$ after dynamic expansion; varying $|A| \in \{40, 70, 100\}$ produced no material change in single-seed checks). Illegal actions (e.g. division by zero) are masked at each step (Appendix A.3).

*Rewards.* Define the *complexity $C$* of an equation as the total number of nodes and edges in the expression tree.[1] For example, $C(ax + b) = 5 + 4 = 9$ and $C(ax^2 + bx + c) = 10 + 9 = 19$. The reward is then

$$
R(\text{action}) = C(\text{equation}) - C(\text{equation after action}) \tag{6}
$$

i.e., reward actions that simplify the equation.

**State Transition Function**. We use SymPy Meurer et al. (2017) to apply operations to a tracked $(lhs, rhs)$ pair. For example, from $(ax + b, 0)$, $(\text{sub}, b)$ gives $(ax, -b)$ and then $(\text{div}, a)$ gives $(x, -b/a)$. The episode terminates when $lhs = x$ and substituting the $rhs$ into the current tracked equation simplifies to 0; after CoV or constant relabeling, the reported answer is mapped back by exact composition of the stored inverse transformations.

**Solving criterion, domain assumptions, and trace validity.** We now state precisely what "solved" means, since the paper's claims should be read against this benchmark-specific criterion rather than against complete symbolic solving.

*One terminally verified root, not the solution set.* An episode counts as solved when three conditions hold simultaneously: (i) the tracked $lhs$ is exactly the solve variable $x$; (ii) the $rhs$ contains no occurrence of $x$; and (iii) substituting the candidate $rhs$ for $x$ into the current tracked equation symbolically reduces to 0 under a fixed hierarchy of SymPy canonicalizers (`expand`, `powdenest`, `ratsimp`, `simplify`, applied independently with operation-count guards). In an episode with no CoV, the tracked equation is the original equation (up to any exactly inverted constant relabeling). With CoV active, the check is performed in the active, post-substitution coordinate system and the verified root is then mapped to the original variable by exact

---

[1] We judged number of nodes+edges correlates better with algebraic complexity than number of nodes alone.

composition of the stored inverse substitutions; we do not run a second independent simplification check against the original pre-CoV expression. The target is thus *one satisfying root under this terminal criterion*: multi-root equations count as solved when any one verified root is produced (e.g. $\sin x = 0$ at $x = 0$), and we make no claim about recovering complete solution sets or about equivalence-preserving derivations.

*Domain and branches.* All coefficients are symbolic and treated as real; sqrt, log, and the inverse trigonometric operations use SymPy's principal branch. An action whose principal-branch result drops the tracked root (e.g. asin applied when the solution lies on another branch) simply produces a state from which the terminal check cannot pass – the episode fails rather than emitting a wrong answer.

*Extraneous and lost roots.* Individual actions need not be equivalence-preserving: square can introduce extraneous roots, and division can remove roots. Extraneous candidates that fail the active-equation terminal check are rejected. Lost roots are handled in two parts: divisions by factors that vanish at a root of the equation are masked from the action set (Appendix A.3), and because success requires only one root, losing a *different* root does not compromise the criterion.

*What is verified, and when.* Intermediate steps are legal algebraic operations applied to both sides via exact symbolic (SymPy) transitions, but are *not* individually checked for equivalence; the terminal criterion is enforced by the active-equation substitution check above. The trace is therefore replayable and inspectable – every step is an explicit symbolic operation – but it is not a proof object certifying every intermediate state equivalent to the original equation. Verification failures of any kind – including SymPy exceptions or canonicalizer op-count limits being exceeded – are conservatively treated as *not solved*.

**Closed vs. open equations**. The MDP formulation above only handles equations that are *closed*, in the sense that solving them requires manipulating terms already present in the equation/sub-expression list. By contrast, solving *open* equations requires adding new, out-of-equation terms or clever substitutions. A classic example is the quadratic equation $ax^2 + bx + c = 0$ which is solved by completing the square – adding $(b/2a)^2$ to each side. We call this reasoning *out-of-term-set*, since the term $(b/2a)^2$ is *not* in the term set we have defined; "generative" here means out-of-term-set, not open-ended expression synthesis. Equations that require such out-of-term-set actions are beyond the scope of the closed-action formulation just described, and were not studied in prior RL work either Poesia et al. (2021); Dabelow & Ueda (2024). We extend the formulation to the open setting in Section 4.2 by exposing the change-of-variables as a learned macroaction inside the same dynamic action space.

### 3.1 TreeMLP: a tree-structured policy network

We introduce **TreeMLP**, a lightweight graph-based policy/value network purpose-built for symbolic expression trees. Nodes are tokens (operators, symbols, or terminals) and edges are parent→child relations. Each node ID is embedded and linearly projected to a hidden size, then $K$ rounds of message passing update the node states with a strictly-local sum aggregator over incoming edges (no degree normalisation): at round $t$,

$$h_i^{(t+1)} = \text{MLP}\left(\left[\text{emb}_i \,\|\, \sum_{j \to i} h_j^{(t)}\right]\right), \tag{7}$$

where the MLP is two ReLU layers and $\text{emb}_i$ is the static node embedding. Padding-aware masks zero out invalid nodes and edges at every stage. After $K$ iterations a fixed-size graph feature is produced by masked mean (or max) pooling over valid nodes and fed to the PPO policy and value heads.

These choices – direct node-ID embedding, sum aggregation without degree normalisation, no learned edge transforms, masked pooling – align TreeMLP with the rooted, sparse structure of expression trees and avoid two encoder pathologies on this domain: GCN's Kipf & Welling (2017) degree normalisation dilutes a single parent's message when an operator has many children (e.g. add with 5 summands), and GraphSAGE's Hamilton et al. (2017) learned edge transforms overfit the small algebraic vocabulary. TreeMLP outperforms both at equal hidden size on the closed-equation small dataset (Appendix A.10; Figure 4) and trains stably with PPO.

The full architecture diagram, message-passing pseudocode, and ablations against vanilla MLP / GCN / GraphSAGE baselines are in Appendix A.10.

## 3.2 Inductive biases

On top of TreeMLP we use three additional inductive biases.

**Curriculum learning.** The agent solves a single equation chosen randomly from a larger set each episode. Equations are selected via *inverse sampling*: the probability of choosing an equation is inversely proportional to the number of times it has been solved before, so the agent spends more time on equations it has not yet mastered.

**Success-replay buffer.** Solution traces in our environment are *exact*: once a sequence of (op, term) actions solves an instance (e.g., $ax+b = 0$), replaying the same actions deterministically reproduces the solution. We exploit this with a light-weight success-replay path: solution traces $(s_t, a_t)_t$ are stored to a memory buffer and periodically mixed into the on-policy rollouts.

**Relabel-constants macroaction.** To improve parameter sharing across structurally identical equations that differ only by symbol names, the agent has a *relabel-constants* action that applies a consistent partial renaming (e.g., $\{d \mapsto a,\ e \mapsto b,\ \dots\}$) to the current state, mapping arbitrary coefficient symbols onto a canonical alphabet $\{a, b, c\}$. The environment stores the substitution stack and, upon solving, unwinds it so the answer is expressed in the user's original symbols. This reduces input vocabulary entropy, prevents overfitting to symbol IDs, and makes action terms like "subtract $b$" reusable across many instances.

**Curiosity bonuses (negative result).** Curiosity bonuses (ICM, RND, NGU) fail to match the no-curiosity baseline on top of TreeMLP; details and a plausible explanation are in Appendix A.13.

## 4 Results

### 4.1 Closed equations

*Algorithms and training.* We use Stable-Baselines3's Raffin et al. (2021) PPO Schulman et al. (2017) (A2C/DQN were worse in initial experiments); hyperparameters are in Appendix A. Each run is $10^7$ steps with evaluation every $5 \times 10^5$, over three random seeds.

*Datasets.* We construct two recursive-template datasets by starting from $x$ and applying random (operation, term) actions, with terms restricted to $\{a, b, c\}$ (to favour long over wide equations). We discard equations SymPy cannot solve and count multi-root cases solved when any one root is found (e.g. $\sin x = 0$ at $x = 0$). `small` contains all depth-$<4$ equations (3874, subsampled $10^3/10^2$ train/test); `large` contains all depth-$<5$ ones (15625, subsampled $10^4/10^3$). We additionally use the CommonCore `poesia` benchmark.

*Results.* Table 1 reports end-of-training greedy test accuracy for the inductive-bias ablation. On the `poesia` CommonCore benchmark, PPO-TREE-RC-BUF reaches 0.93 greedy test accuracy on the full benchmark, matching ConPoLe's 0.925 Poesia et al. (2021) under matched greedy decoding. Restricting to non-degenerate templates (excluding edge cases the closed-action set cannot reach) raises accuracy further – to 0.955 for PPO-TREE-RC and $\geq 0.98$ on the solvable subset (Appendix A.5). Each bias contributes: TreeMLP alone lifts `poesia` from 0.17 (vanilla MLP) to 0.80; relabel-constants adds a further jump on `small`; and the success-replay buffer is critical for scaling – PPO-TREE-RC-BUF degrades least with depth ($0.85 \to 0.50$ `small`→`large`, vs $0.85 \to 0.10$ for PPO-TREE-RC).

*Learned representations.* t-SNE van der Maaten & Hinton (2008) projections of the TreeMLP pooled embeddings cluster cleanly by equation family (linear, radical, trig, log occupy disjoint regions), indicating the encoder captures structural identity rather than surface tokens (Appendix A.13, Figure 8).

### 4.2 Open equations

We now consider the harder class of *open equations*: those that cannot be solved by rearranging the sub-expressions already present in the equation, but require introducing a new term via a *change of variables*

Table 1: Greedy test accuracy on the three closed-equation benchmarks for the inductive-bias ablation. ConPoLe / ConPoLe-local numbers from Poesia et al. (2021). `small`/`large` are our depth-$<4$ / depth-$<5$ internal benchmarks; `poesia` is the CommonCore equations split of Poesia et al. (2021).

| Algorithm | small | large | poesia |
|---|---|---|---|
| ConPoLe-local | n/a | n/a | 0.765 |
| ConPoLe | n/a | n/a | 0.925 |
| ppo | 0.04 | 0.02 | 0.17 |
| ppo-tree | 0.25 | 0.10 | 0.80 |
| ppo-tree-rc | 0.85 | 0.10 | 0.85 |
| ppo-tree-buf | 0.55 | 0.40 | 0.85 |
| **ppo-tree-rc-buf** | **0.85** | **0.50** | **0.93** |

(CoV) – a substitution $x = \phi(y)$ that re-expresses the equation in a new variable $y$. We consider four template families, each paired with its solving substitution:

$$ax^2 + bx + c = 0, \quad x = y - b/(2a) \tag{8}$$

$$ax^3 + bx^2 + cx + d = 0, \quad x = y - b/(3a) \tag{9}$$

$$ax^4 + bx^3 + cx^2 + dx + e = 0, \quad x = y - b/(4a) \tag{10}$$

$$ae^{kx} + be^{-kx} + c = 0, \quad x = \tfrac{1}{k} \log y \tag{11}$$

Each substitution depresses (or, for the exponential, rationalizes) the equation, after which standard closed-action steps finish the solution. The polynomial families are restricted to fully-depressable special forms: e.g. the quartic dataset constrains $(c, d, e)$ to functions of $(a, b)$ so that $y = x + b/(4a)$ reduces the equation to $Ay^4 + B = 0$, solved by two applications of $\sqrt{\cdot}$; generic quartics, which admit no closed-form depression by a fixed CoV, are excluded. The exponential case yields a Laurent polynomial in $y$ that resolves to a quadratic after one further CoV (exponential block of Table 10). We expose the CoV as a single *macroaction* in the same dynamic action space: when the policy selects it, the substitution $\phi$ is generated by a learned policy $\pi_{\text{cov}}$ (Section 4.2.5) and applied to the equation. The main policy selects the trigger and composes it with ordinary steps, while $\pi_{\text{cov}}$ supplies the substitution; Section 4.2.3 tests which trigger decisions are genuinely non-trivial. This keeps the open problem inside the same MDP, with one policy trained on a mixed closed/open dataset (Section 4.2.1).

**Benchmark scope.** We call this the *Restricted-Open* setting: arbitrary same-degree polynomials with non-special coefficients (e.g. a generic $ax^4 + bx^3 + cx^2 + dx + e$) are out of scope, as they admit no closed-form depression by a fixed CoV. The goal is to study whether RL can learn *when* and *how* to invoke a CoV macroaction in a controlled regime where one exists; extending to fully open families is future work.

### 4.2.1 Method

We train a single TreeMLP-PPO policy on mixed datasets of closed and open equations: `open_small` (916 train / 91 test, stratified across the four CoV families and the `abel_level1/2/3` closed classes) and a 10×-larger `open_large`. (`abel_level1/2/3` are three closed-equation difficulty tiers of increasing tree depth and coefficient structure, `abel_level3` the hardest; throughout, *coverage* denotes the fraction of *training*-set equations the agent solves, as distinct from held-out test accuracy.) The policy carries over the closed-equation inductive biases – curriculum learning, constant relabeling, and a success-replay buffer (Section 4.1) – with four additions specific to the open setting, which we state up front and ablate below.

**Action set additions.** The open setting adds (i) a cube-root primitive (needed because the depressed-cubic CoV collapses an equation to $ay^3 + k = 0$ which $\sqrt{\cdot}$ alone cannot finish) and (ii) the CoV macroaction itself, with a per-episode cap of 3 applications to permit nested substitutions (an exponential equation first becomes a rational form, then a quadratic).

**Action-diversity penalty.** A small reward penalty ($\alpha = 0.1$) is subtracted whenever the current action's integer index matches the previous step's – exact (operation, term) identity, not operation alone. The penalty

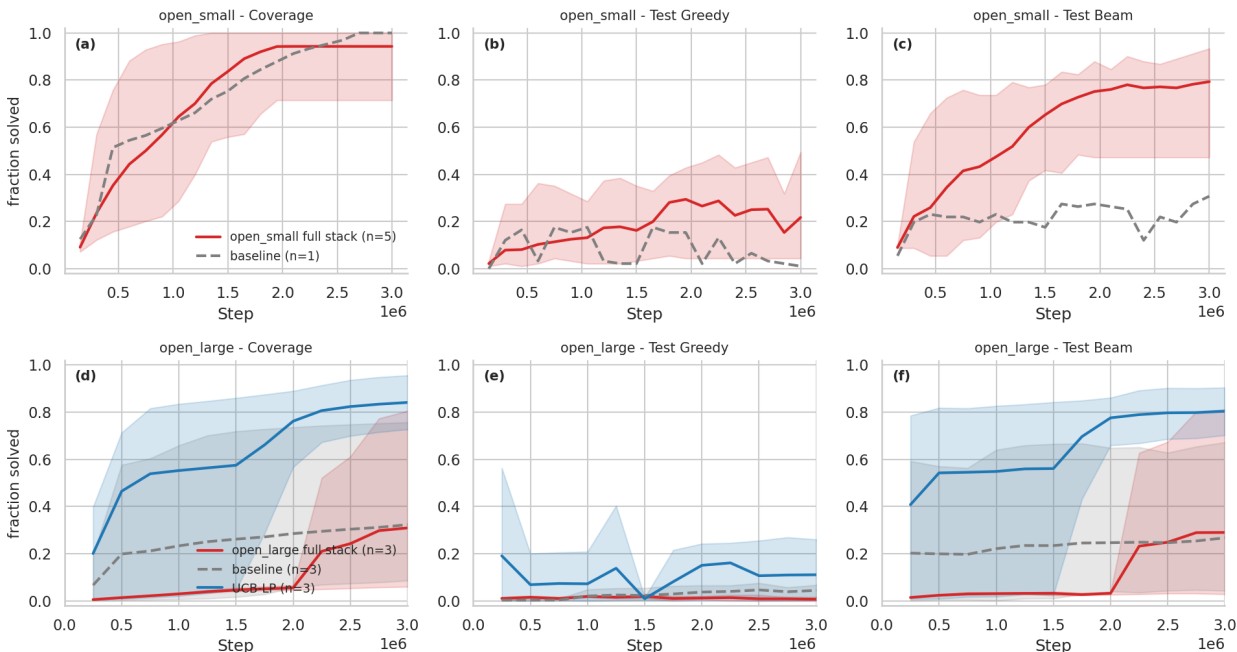

Figure 1: **Open-equation learning curves** (coverage, greedy and beam-width-5 test accuracy) for `open_small` (top) and `open_large` (bottom). Red: full method stack; grey dashed: baseline; blue: UCB-LP. Solid lines are seed means, bands min/max; seed counts per cohort are given in the text. Per-seed distributions are in Figure 3.

targets short-cycle loops: 93% of unsolved `abel_level3` failures on the larger benchmark (Section 4.2.4) are 1-cycles (81%, caught by the penalty) or $a \to b \to a$ 2-cycles (12%, slipping past a one-step penalty; we return to this in Section 6).

**Freshness-managed success replay.** The buffer discards its oldest half every 20 rollouts. The design goal is to keep the behaviour-cloning target on the current frontier rather than on stale successes; however, the seed-matched ablation in Table 11 finds that its incremental effect changes sign across seeds, so we make no empirical improvement claim for this component.

**Value-guided beam search.** At evaluation we score beam paths by

$$\text{score(path)} = \sum_t \log \pi_\theta(a_t \mid s_t) + \lambda V_\theta(s_T) - \beta C(s_T), \tag{12}$$

combining the trained policy log-prob with a value-head look-ahead ($V_\theta$) and an equation-complexity proxy ($C$). We use width 5 throughout ($\lambda = \beta = 0$ recovers plain beam). The headline `open_small` beam numbers and the seed-matched ablation (Table 11) are decoded at $\lambda = 0$; the per-class tables (Tables 13, 14) at $(\lambda, \beta) = (1, 0)$. The two differ little — on our 3M checkpoints, switching $\lambda$ moves $test_{\text{beam}}$ by at most 0.044 (mean 0.023) — but we label each table rather than assert equivalence; varying widths $\{3, 5, 7, 9\}$ and $\lambda \in \{0, 1\}$ produced no material change at convergence (observed spread $\leq 0.025$, no table). Beam entries reaching the same canonical equation are deduplicated.

### 4.2.2 Results

Unless stated otherwise, checkpoints are selected by validation beam on a disjoint 30% validation slice and reported on the remaining 70% test slice (Section A.14); $test_{\text{beam}}$ is value-guided beam width 5 (Eq. 12); greedy is top-1 decoding of the same checkpoint.

A plain PPO-TREE-RC-BUF-COV agent on `open_small` reaches $coverage = 1.0$ on the training set but only $test_{\text{beam}} = 0.31$ on test (`open_small`, $n=1$): it memorizes a single CoV recipe and fails to generalize across

the four families. Adding the four open-equation ingredients above and selecting the best checkpoint by validation accuracy (Section A.14), we obtain a 5-seed mean $test_{\text{beam}} = 0.79$ (`open_small`, $n = 5$, range 0.47–0.93). Its seed-matched baseline is $0.354 \pm 0.090$ ($n = 3$; the $n = 1$ figure of 0.31 is superseded). The corresponding 5-seed greedy (top-1) mean is 0.67 (range 0.40–0.80), so value-guided beam lifts the greedy policy by $\approx 0.12$ rather than supplying the bulk of the accuracy (the three-way greedy/beam/search comparison is in Table 2 and the beam-width ablation in Figure 2). One of the five seeds early-stopped at 1.95M of 3M steps before fully learning the CoV classes ($test_{\text{beam}} = 0.47$), and is included in the 5-seed mean as reported. Figure 1 (top row) shows the learning curves with mean and min-to-max band across the five seeds.

**Ablation (seed-matched).** All four training configurations on the *same* three seeds, same 3M-step budget, decoder fixed, no checkpoint selection (Table 11). The CoV machinery (cube-root + nested CoV) carries essentially the whole effect: $+0.594$, positive on every seed. The action-diversity penalty is **harmful** ($-0.245$, negative on every seed). The freshness-managed buffer is unresolvable (sign changes across seeds). Removing the penalty ($\alpha = 0$) raises the full-stack control from 0.490 to 0.891 and gives end-to-end lifts of $+0.521/+0.537$, both positive on every seed, with 0/9 stalls. We retain 0.79 as the headline because all downstream analyses used the $\alpha = 0.1$ cohort. The `open_large` cohorts (Section 4.2.4) were run at $\alpha = 0$, so the bimodality there is not an artefact of this penalty.

**Per-class accuracy.** On `open_small` the full stack solves all four CoV classes at 88–100%, lifting the exponential class from 0% to 88%; `abel_level3` remains weakest (0.53, full stack, 4-seed mean), its failures dominated by short action cycles (breakdown in Table 13). On `open_large` the pattern is sharper: across escape seeds the CoV classes saturate ($\sim 0.98$), `abel_level3` stays weak (escape-seed mean 0.39, range $[0.07, 0.55]$; Table 14).

**Solution trace: nested change-of-variables.** The exponential class is the only one of the four CoV families that requires *two* CoV steps; the agent discovers this nested recipe on its own (full greedy trace in the exponential block of Table 10, Appendix A.8). The policy first applies CoV ($x \to \log x/k$) to turn the exponential equation into a rational one, expands to a quadratic, then invokes CoV *again* ($x \to x - B/(2A)$) to depress the quadratic. The same agent solves quadratic, cubic, and quartic classes with a single-CoV recipe.

**Policy vs. search.** Table 2 compares three decoders of the same policy against two non-learned searches. Greedy top-1 alone reaches 0.67 – above BFS (0.26) and A$^*$ (0.38–0.64, same $10^5$-expansion budget). Value-guided beam adds $\approx 0.12$ to reach 0.79 at $\leq 250$ SymPy transitions per equation. The accuracy is policy-dominated: beam contributes a modest margin, not the bulk of the result. No learned external baseline exists for the restricted-open setting; ConPoLe and Dabelow & Ueda (2024) do not transfer (see Related Work).

| decoder / search | policy? | search? | `open_small` acc. |
|---|:---:|:---:|:---:|
| Greedy (top-1 policy rollout) | ✓ | | 0.67 (0.40–0.80) |
| Value-guided beam (width 5) | ✓ | ✓ | **0.79** (0.47–0.93) |
| BFS ($N = 10^5$ expansions) | | ✓ | 0.26 |
| A$^*$ (complexity heuristic, $10^5$) | | ✓ | 0.38–0.64 |
| SymPy oracle | – | – | 1.00 |

Table 2: **Policy vs. search on `open_small`** (5-seed cohort; ranges are min–max over seeds where applicable). The learned policy's greedy top-1 rollout (0.67) alone exceeds the strongest non-learned search over the same action space (A$^*$ up to 0.64, BFS 0.26); value-guided beam adds a further $\approx 0.12$ to 0.79. The accuracy is thus policy-dominated: search contributes a modest margin, not the bulk of the result. The SymPy oracle upper-bounds the benchmark.

**Decomposing the open-equation pipeline.** We re-evaluate the converged `open_large` agent (seed109, `open_large`, $n = 1$; $test_{\text{beam}} = 0.81$ on the full 1,000-equation held-out file) in three configurations (Table 3): (A) the full learned pipeline; (B) the CoV macroaction *disabled* (the agent must solve using only closed-action steps); and (C) an *oracle CoV* (SymPy's depression substitution, applied on every episode in which the agent

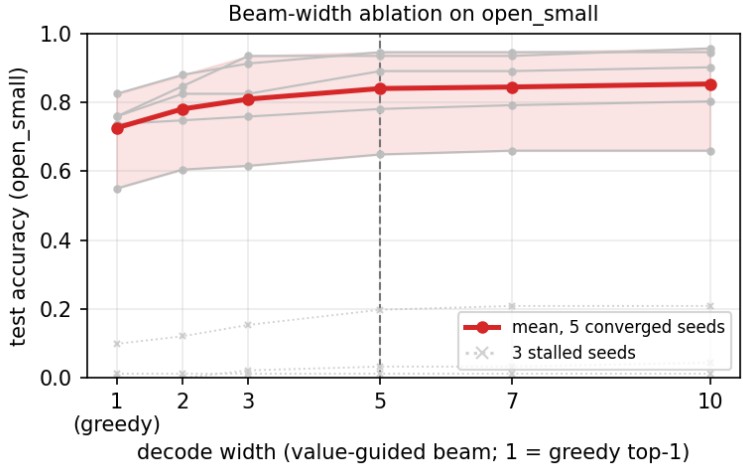

Figure 2: **Beam-width ablation on `open_small`.** Value-guided-beam test accuracy vs. decode width (width 1 = greedy top-1) across eight full-stack `open_small` checkpoints; solid grey lines are the five converged seeds, dotted the three stalled seeds (greedy ≈0, consistent with the seed-level bimodality of Section 4.2.4), red is the mean over the converged seeds with min–max band. On the converged seeds accuracy rises from greedy (0.73) through width 5 (0.84) and then saturates, supporting width 5 as the reported operating point; a stalled policy solves ≈0 at every width, so beam-width is only meaningful once the policy has learned solution paths. (These checkpoints were trained to 5M steps and sit above the 3M-step headline cohort in absolute level; the width-dependence *shape* and the high greedy anchor are the point.)

invoked the CoV macroaction — 818 of the 1,000 held-out episodes; note this exceeds the 800 CoV-*family* equations, because the agent also invokes CoV productively on some closed equations, cf. Section 4.2.3). All denominators are the full 1,000-equation held-out file. Configuration B does *not* show CoV is a train-from-scratch necessity: the trained policy simply does not degrade gracefully when CoV is masked at test time, a regime it was never trained for. Configuration C isolates substitution *generation*: swapping $\pi_{\mathrm{cov}}$ for SymPy's exact depression moves accuracy by ≤0.02 across escape seeds (the decomposition seed `seed109`: $0.81 \to 0.799$; three further escape seeds, $A \to C$: $0.79 \to 0.80$, $0.90 \to 0.90$, $0.91 \to 0.90$), so the open-equation accuracy comes from the CoV-trigger and closed-action policy, not $\pi_{\mathrm{cov}}$'s decoder.

Table 3: **Pipeline decomposition** on `open_large` (seed109; beam width 5; denominator = 1,000 held-out equations). Config B masks the CoV macroaction at test time; config C substitutes SymPy's exact depression for $\pi_{\mathrm{cov}}$ on the 818 episodes in which the agent invoked CoV.

| config | what changes | beam acc |
|---|---|---|
| A | full learned pipeline | 0.81 |
| B | CoV macroaction masked at test | 0.004 (4/1000) |
| C | oracle (SymPy) CoV vs. $\pi_{\mathrm{cov}}$ | 0.799 (799/1000) |

### 4.2.3 Is the CoV trigger learned, or would a rule do?

A central question is whether the policy has recovered a non-trivial decision about *when* to invoke a CoV. We therefore test the trigger directly against hand-coded alternatives. We hold the trained policy fixed and *mask* the CoV action from it, handing the trigger decision to an external rule; the policy still selects every non-CoV action. Four regimes, all greedy-decoded on the same checkpoints:

- **never** — CoV masked for the whole episode.

- **always@0** — fire CoV unconditionally as the first action, then hand over to the policy. Isolates *detection* from *timing*.

- **rule** — fire CoV iff the depression detector returns a non-identity substitution on the current form and the CoV budget is unspent. This is the natural rule: fire whenever a CoV is available.

- **rule+expand** — as **rule**, but fire only once the tracked lhs is already expanded ($\text{expand}(\ell) = \ell$); when a CoV is detectable on an *unexpanded* form, the rule spends one forced `expand` step first. This regime is therefore strictly stronger than a trigger: it overrides the policy on one non-CoV action as well. We report it because it is the rule that works, and we flag the extra power it is given rather than presenting it as trigger-only.

All regimes use the *same* substitution generator as the learned agent, and (with the single exception just noted) delegate every non-CoV action to the trained policy, so the comparison isolates the trigger.

Table 4: **CoV trigger ablation.** Greedy end-task accuracy on `open_small` (5 converged seeds; mean with min–max) and `open_large` (seed109). The policy is fixed and identical across rows; what changes is *who decides when to fire CoV* (and, in the last row only, one forced `expand` step — see Section 4.2.3). Masking CoV drives the open classes to zero: for *this trained policy* the macroaction is load-bearing, though as with config B of Table 3 this does not establish that CoV is necessary to *learn* the task, since the policy was never trained for the masked regime. Firing CoV blindly at $t\!=\!0$ recovers most, but not all, of the open accuracy, so *timing* matters. The naive rule does not beat blind firing on open equations — it fires at $t\!=\!0$ on every one of them (Table 5 explains why). *Denominators*: the full `open_small` test file ($n\!=\!91$: 60 CoV-requiring + 31 closed) and the full 1,000-equation `open_large` file; every row shares the same denominator, so the comparison is internal and like-for-like. The cohort is the eight `open_small` checkpoints of Figure 2 (5M steps, evaluated on the same 91-equation file; the 5 converged seeds, with the 3 stalled seeds reported separately in Appendix A.13 and never pooled), so the **learned** row reproduces that figure's greedy anchor (0.73) exactly — which is our correctness gate on the harness. These are greedy numbers on the 5M-step cohort and are not comparable to the beam numbers of Section 4.2.2, which use a different decoder and the 3M-step headline cohort.

| | open_small (5 seeds) | | | open_large |
| --- | --- | --- | --- | --- |
| CoV trigger | all | open | closed | all (seed109) |
| never | 0.08 [0.05, 0.12] | 0.00 | 0.25 | 0.004 |
| always@0 | 0.57 [0.54, 0.60] | 0.73 | 0.25 | 0.602 |
| rule (detector only) | 0.62 [0.54, 0.68] | 0.73 | 0.41 | 0.603 |
| **learned (ours)** | 0.73 [0.55, 0.82] | **0.90** | 0.40 | 0.771 |
| rule + expand | 0.75 [0.62, 0.82] | 0.92 | 0.41 | 0.787 |

**What the aggregate hides.** In aggregate the learned trigger sits between the naive rule (0.62) and the expand-preconditioned rule (0.75), and the latter edges it out. The per-family breakdown (Table 5) shows this aggregate is misleading in both directions, and locates the entire trigger question in one family.

**The trigger question is the exponential family.** On quadratic, cubic, and quartic the CoV is detectable in the initial state and firing it immediately is optimal; **rule** therefore fires at $t\!=\!0$ on every open equation (its first-CoV step is 0 in 60/60 cases), which is why it collapses onto **always@0** on the open classes. There is nothing to learn here, and we say so: for single-CoV families a one-line detector is as good as the policy.

The exponential family is different. It is the only family requiring *two* CoVs ($x\!\rightarrow\!\log x/k$ turns the equation into a rational one, which must then be expanded to a quadratic and depressed; Section 4.2.2). The naive rule solves 0 **of** 75. The failure is not detection but *timing*: at the second CoV the tracked lhs is the unexpanded product $x\,(-3a/x - 5cx + 5e)$. The detector reports a quadratic (it matches on $\text{expand}(\ell - r)$, which is $-3a - 5cx^2 + 5ex$, and returns $x \mapsto x + e/(2c)$) but the substitution is applied to the *raw* form, so $-3a/x$ becomes $-3a/(x + e/(2c))$), which never cancels; the episode ends in a rational form the closed actions cannot finish. Expanding first instead yields the clean depressed quadratic $-3a - 5cx^2 + 5e^2/(4c)$.

Table 5: **Per-family trigger ablation**, open (CoV-requiring) equations only, `open_small`, 5 converged seeds (75 = 15 equations × 5 seeds per cell). On the three polynomial families the trigger is *trivial* — the CoV is detectable at $t=0$ and firing immediately is optimal — and every rule *matches or beats* the policy (the policy is 2 behind on cubic and 5 behind on quartic, where it occasionally declines to fire). The exponential family is the only one requiring a *nested* CoV, and it is the only one where the trigger decision has any content: the naive rule scores **zero**.

| family | never | always@0 | rule | **learned** | rule + expand |
|---|---|---|---|---|---|
| quadratic | 0/75 | 70/75 | 70/75 | 70/75 | 70/75 |
| cubic | 0/75 | 75/75 | 75/75 | 73/75 | 75/75 |
| quartic | 0/75 | 75/75 | 75/75 | 70/75 | 75/75 |
| exponential | 0/75 | **0/75** | **0/75** | **56/75** | 57/75 |

The learned policy does exactly that — expands, then fires — solving 56/75 from reward alone. Adding that precondition to the rule (**rule+expand**) recovers 57/75 and closes the gap.

**What we conclude.** The learned trigger is matched — narrowly exceeded, by $\approx 0.02$ — by a hand-coded rule, and only by a rule carrying a precondition we obtained by inspecting the learned policy's traces. We therefore state the contribution as *recovery* rather than superiority: the policy recovers, from reward alone, a trigger as good as a hand-engineered detector, without being given the detector, and it is the only method here that solves the nested-CoV family without being told how. Where a closed-form depression detector exists per family, a rule suffices and should be used; the learned trigger's value is that it requires no such detector, which is what makes the formulation portable to settings (e.g. differential-equation reductions) where no per-family detector is available. We regard this as the honest scope of the "learns *when*" claim.

**Trigger accuracy.** As a binary CoV-required classifier over the full $n=91$ file (5 converged seeds): precision 0.90, recall 0.97, $F_1$ 0.93. Recall is near-ceiling; the 0.23 false-positive rate on closed equations is partly benign — 71% of those firings occur on episodes the policy goes on to solve (e.g. squaring a closed equation to produce a solvable quadratic).

### 4.2.4 Scaling to `open_large`: a bimodal failure mode and a candidate mitigation

**The bimodality.** On the 10×-larger `open_large` benchmark (10,000 train / 1,000 held-out equations, same composition, split 30/70 into validation/test), the agent exhibits a sharp seed-level *bimodality*. Across all cohorts (Table 6; full sweep in Table 15), results partition cleanly into two regimes with no seed landing in $(0.10, 0.27)$: an *escape* regime ($test_{\text{beam}} \geq 0.2$, observed range $[0.27, 0.91]$) and a *stall* regime ($test_{\text{beam}} \leq 0.10$). The same code, data, and hyperparameters produce both regimes; only the seed differs. This resembles bimodal-across-seeds capability acquisition in LM pretraining Zhao et al. (2025) and the *policy confounding* Suau et al. (2024) / *capability boundary collapse* Dong et al. (2025) failure modes.

**Mechanism: per-class trap dynamics.** A post-hoc analysis of in-training solve events classifies each successful step by the source class of the equation it solved. Escape seeds distribute their solves roughly uniformly across the four CoV classes (20–25% each) and `abel_level3` ($\sim 10\%$). Stall seeds concentrate $\geq 85\%$ of their solves on `abel_level3`, the easiest class *during training*, and approach zero on the CoV classes. Stalled seeds *did* learn – they routinely solve training `abel_level3` instances – but fall into short action cycles on held-out instances. **Escape correlates with class diversity in training events; stall corresponds to `abel_level3` lock-in**, via the short-cycle failure pattern quantified in Section 4.2.1 (93% of unsolved cases are 1- or 2-cycles).

Across three probes we find *no evidence* for the three most natural alternative explanations: *capacity loss / dormant neurons* Sokar et al. (2023) (0% dormant on both stuck and escape seeds at the standard ReDo threshold), *data-distribution skew* (an `abel_level3`-skewed training set transfers as 0.04 to the standard split), and *encoder representation collapse* (t-SNE clusters cleanly on stalled seeds). Details in Appendix A.13. These probes are necessary-not-sufficient (a clean encoder does not rule out a head-level issue), but the evidence is most consistent with a *policy-attractor* phenomenon.

Table 6: Intervention sweep on `open_large`. We report test beam at the validation-selected checkpoint for each seed (checkpoint chosen by val beam on the 30% split; test beam on the disjoint 70% split). Escape $= beam \geq 0.2$. Mean is reported both over all seeds (mixing escape and stall) and over escape seeds only, since the bimodal regime makes the all-seeds mean a noisy summary. *Crash convention:* the two UCB-LP seeds that crashed on a SymPy/mpmath overflow were already stalled pre-crash; we count them as stalls (6/8), and parenthetically report the figure excluding them (6/6). The Fisher/Wilson statistics on these counts are in the text. Headline cohorts only; full sweep in Appendix A.13. LP follows Matiisen et al. (2019); oracle = SymPy on the same test set.

| Cohort | $n$ | Escape rate | Mean (all) | Mean (escape) | [min, max] escape |
|---|---|---|---|---|---|
| Baseline (full stack) | 5 | 3/5 | 0.30 | 0.47 | $[0.27, 0.83]$ |
| LP curriculum (vanilla) | 3 | 2/3 | 0.40 | 0.58 | – |
| **UCB-LP curriculum** | **8** | **6/8** (6/6) | **0.63** | **0.83** | $[0.69, 0.91]$ |
| Oracle (SymPy) | – | – | 1.00 | 1.00 | – |

**Intervention sweep.** We evaluated seven intervention cohorts targeting the failure mode; Table 6 reports the headline cohorts (baseline, vanilla LP curriculum, UCB-LP) with the full 9-row sweep in Appendix A.13. The most effective is a *UCB-style learning-progress curriculum* (UCB-LP), which replaces vanilla LP's Matiisen et al. (2019) heuristic exploration floor with a UCB1 Auer et al. (2002) bonus on per-class sampling weights: $w_c \propto \hat{p}_c + \beta\sqrt{\log T/n_c}$ ($\hat{p}_c$ = EMA learning progress on class $c$, $n_c$ its visit count, $\beta = 2.0$), structurally similar to ACTOR-CURATOR Gu et al. (2026). Across $n = 8$ training runs it achieves 6/8 escape with mean $beam = 0.83$ over escape seeds (range 0.69–0.91); the best UCB-LP seed reaches 0.911 (+0.10 over the previous single-seed best of 0.81). The escape-rate change from 3/5 (baseline) to 6/8 (UCB-LP) is *not* statistically significant (Fisher exact $p \approx 1.0$; Wilson CIs overlap); a powered claim would need $\geq 20$ seeds per cohort. UCB-LP should therefore be read as a candidate mitigation direction, not a demonstrated fix. The two non-escape seeds crashed on a SymPy/mpmath `OverflowError` while already stalled; we count them as stalls (6/8), giving 6/6 if excluded. UCB-LP was selected for expansion to $n = 8$ because it won the initial 3-seed screen, so cross-cohort comparisons are exploratory. All numbers are val-best per seed on a 30/70 split (Appendix A.14).

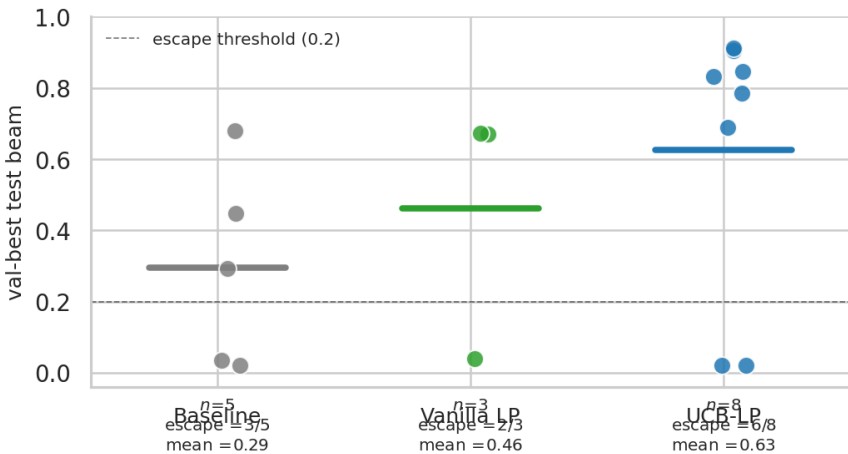

Figure 3: **Per-seed val-best test beam on `open_large`.** Each dot is one training run; horizontal bar is the cohort mean. The escape mode (beam $\geq 0.2$; escapes cluster near 0.7–0.9) is more populated and reaches higher values under UCB-LP, while the stall mode at $\sim 0.02$ is preserved (UCB-LP: 2/8 stalls, both from crashed seeds whose pre-crash trajectories were already stalled).

### 4.2.5 Learning the change of variables

$\pi_{\text{cov}}$ maps an equation to its depression substitution, trained by supervised learning on the known closed-form targets (grammar and worked example in Appendix A.7). It achieves 100% on 192 held-out test equations across eight seeds under greedy decoding, generalizing to renamed coefficients within the four training families; generalization to new substitution structures or template families outside the training grammar is untested and is future work. As shown in Section 4.2.2 (config C), it is interchangeable with a CAS depression call and is not load-bearing for end-task accuracy; the RL problem isolates *when* and *how* the policy invokes CoV, not the substitution generator.

## 5 Conclusion

We presented the first reinforcement-learning agent to solve *restricted-open* (change-of-variables) symbolic equations step-by-step with the main policy trained from reward alone (the CoV substitution generator is supervised but interchangeable with a CAS call), extending RL equation solving from the nonlinear *closed* case (O'Keeffe, 2025) to the open setting – recovering a CoV-conditioned procedure, including trigger timing where it is non-trivial, and composing CoV with ordinary algebraic steps under a single policy. The trigger ablation (Section 4.2.3) narrows the claim to *recovery*: for single-CoV families a one-line rule suffices; the trigger has content only for the nested-CoV exponential family, where the natural rule solves 0/75 and the policy solves 56/75 from reward alone — matched by a rule only once handed a precondition read off the policy's traces. Recovery needs only a reward; the rule needs a known closed-form depression per family, which is the whole difficulty in the ODE reductions this formulation targets. Under the terminal criterion of Section 3, it reaches 0.79 beam-decoded test accuracy (greedy top-1 mean 0.67, itself above all non-learned search baselines, with beam adding a further margin) on restricted-open quadratic, cubic, quartic, and exponential families – beyond the closed-equation setting of prior RL – and matches the prior best on closed CommonCore equations, while exposing a replayable, terminally checked trace that one-shot neural solvers and a bare CAS do not provide. The principal open problems are the gap between decode-time and training-time search (a natural fit for expert iteration) and the seed-level bimodality at $10\times$ scale; no matched $\alpha\!=\!0$ small-scale control cell stalls, while the learning-progress curriculum at $10\times$ shows a non-significant positive trend (a candidate mitigation, not a demonstrated fix). We see the formulation as a step toward RL for the procedural reasoning that differential-equation reductions and prover tactics require.

## 6 Limitations

Several limitations double as concrete next steps.

**Our `open_small` headline cohort carries a reward term that suppresses the matched controls.** Under the seed-matched control the penalty costs $-0.245$ (negative on every seed); removing it ($\alpha\!=\!0$) raises the full-stack control from 0.490 to 0.891 with 0/9 stalls, and end-to-end lifts of $+0.521/+0.537$ positive on every seed (Table 11).

We have not promoted $\alpha\!=\!0$ to the headline because all downstream analyses used $\alpha\!=\!0.1$ checkpoints; a decode-only re-run of the trigger control at $\alpha\!=\!0$ leaves the "recovery, not superiority" finding unchanged (58/75 vs 56/75 on the exponential family). We decline to call 0.79 a lower bound: the seed-matched protocol gives $0.490 \pm 0.377$ with a 1/3 stall rate, so 0.79 may be a favourable seed draw; retraining at $\alpha\!=\!0$ under the headline protocol is the concrete next step.

**The ablation resolves only large effects.** Parallel training is not deterministic at a fixed seed, so seed-matching removes the seed-selection confound but not a residual run-to-run variance (0.079 mean, 0.253 worst case) that no seed count averages away without per-cell replicates. Table 11 can therefore resolve the large sign-stable directions but neither the freshness-buffer increment nor the *penalised* full-stack-versus-baseline contrast, both of which flip sign. A 3/3 sign agreement carries a floor two-sided $p$ of 0.25: these are corroborated directions, not powered tests (Henderson et al., 2018; Agarwal et al., 2021), and we use them to revise no headline number.

**The underlying equations are CAS-solvable.** Every equation we report is solved in milliseconds by SymPy; our contribution is the RL formulation and the learned step-by-step procedure, not the underlying solvability. The framework's value is in downstream settings – differential-equation solving, theorem-prover tactics, didactic solvers – where the trace matters; on bare equation-solving a CAS comparison is unfavourable and would be misleading.

**Residual variance under UCB-LP.** UCB-LP shows a non-significant positive trend (6/8 escape, Fisher $p \approx 1.0$); two seeds still stall. Tighter curricula (e.g. Thompson-sampling class gating Wu et al. (2026)) and a hardened numerical pipeline are the concrete next steps; the 2/8 stall rate and 0.09 gap to oracle are the principal open problems on `open_large`.

**Structural ceilings and the weak class.** The cubic class was unsolvable until we added a cube-root action, and analogous gaps may exist outside our four CoV templates (e.g. quintics, with no radical solution): the action set bounds the solvable class in a way learning cannot overcome. Separately, `abel_level3` stays weakest (0.53 full-stack on `open_small`; escape-seed mean 0.39, range $[0.07, 0.55]$, on `open_large`); its dominant failure is the policy-looping pattern of Section 4.2.4, pointing to reward shaping rather than an action-set gap.

**Search at evaluation, not training.** Value-guided beam adds a further margin over the greedy policy at decode time ($0.67 \rightarrow 0.79$), yet search is applied only at evaluation – the policy trains for greedy rollouts. Folding decode-time search back into training (expert iteration / AlphaZero-style self-improvement Silver et al. (2018)) to close even this residual gap is the natural and most promising direction beyond this paper.

### Broader Impact Statement

This work applies RL to step-by-step symbolic equation solving. The equations are already solvable in milliseconds by computer algebra systems, so the method confers no new equation-solving capability and poses no direct misuse risk; its benefits are indirect, in the explicit, replayable traces it produces for tutoring and for procedural reasoning in ODE solving and prover tactics.

### Reproducibility Statement

The MDP formulation, state/action encoding, reward, and the solving criterion (one terminally checked root; domain and branch conventions; active-equation substitution followed by exact inverse composition) are specified in Section 3; the TreeMLP architecture, pseudocode, and a minimal implementation are in Appendix A.10. PPO and method hyperparameters are tabulated in Appendix A (Tables 7 and 8), and dataset generation is described in Section 4.1 (closed) and Sections 4.2–4.2.5 (open / $\pi_{\text{cov}}$), with the rational-equation construction in Appendix A. The validation-based checkpoint-selection protocol and the 30/70 split (seed 42) are given in Appendix A.14; per-seed counts and the crash-counting convention are stated alongside each cohort. An anonymized repository will accompany the revised submission and will contain: the environment (including the action-mask logic of Appendix A.3 and the terminal-check code), dataset-generation scripts and the exact train/validation/test splits for `small/large/open_small/open_large/cov_large`, the $\pi_{\text{cov}}$ grammar and training data, all training seeds, and the launch scripts we hold for the cohorts in Tables 6 and 15 — with the caveat that for a subset of the earlier cohorts we retain the seeds and recorded metrics but *not* an executable launch script, so their hyperparameters are attested by the run records rather than by a script; those cohorts are marked as such in the release. The seed-matched ablation of Table 11 and the $\alpha = 0$ cells were run from scripts included verbatim. We also release the greedy/beam/BFS/A$^*$ evaluation scripts. Code and datasets will be released publicly upon acceptance.

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

# A Supplementary material

## A.1 Macroactions

Here we justify why the three macroactions (expand, None), (collect, $x$), (mul, $-1$) discussed in the main text are *required* to solve the equations we consider, not mere conveniences for the RL agent. Consider the action set *without* these macroactions:

$$O = \{\text{add, sub, mul, div}\}$$
$$\cup \{\text{square, sqrt, exp, log, sin, cos, asin, acos}\}, \tag{13}$$
$$T = \text{SubExpr(lhs)} \cup \text{SubExpr(rhs)}, \tag{14}$$
$$A = (O \times T). \tag{15}$$

Now consider the equation $dx + c(ax + b) = e$. The agent must distribute $c$ into the bracketed term $ax + b$, which the action set above cannot do: expand is required. Once expanded to $dx + cax + cb = e$, one needs to factor the $x$ common to the first two terms—this is what (collect, $x$) provides. Finally, consider $-x = b$: since $-1$ is not in our term set, (mul, $-1$) is needed to isolate $x$. (Alternatively, $-1$ could be added to the term set, but we chose to keep the term set purely symbolic.)

## A.2 Dataset generation: rational equations

To supplement the recursive datasets, we constructed rational equations of the form

$$\frac{ax + b}{cx + a} + b = 0,$$

where $a, b, c$ are (non-zero) symbolic coefficients. We excluded degenerate cases such as denominators that vanish identically or cancel trivially with numerators.

Examples of retained equations include:

$$\frac{x + b}{cx + a} = 0 \quad \Rightarrow \quad x = -b,$$

$$\frac{ax + b}{cx + a} + b = 0 \quad \Rightarrow \quad x = \frac{-b - ab}{a + cb},$$

$$\frac{ax + b}{cx + a} - c = 0 \quad \Rightarrow \quad x = \frac{-b + ac}{a - c^2}.$$

These rational forms were merged into both the small and large datasets, with their proportion limited to preserve diversity across other functional forms (logarithmic, trigonometric, exponential, etc.).

## A.3 Illegal actions

The main issue was to avoid illegal division by zero. This is not as simple as precluding (div, 0) from the action set. We also had to prohibit *hidden* division by zero—for example, dividing by $(x + a)$ or $(x + b)$ when the equation has the form $(x + a)(x + b) = 0$, since one of the factors will vanish at the solution. To handle this, we check whether the current equation factors as $P(x)Q(x) \cdots = 0$ with $P, Q, \ldots$ polynomial in $x$, and remove (div, $P$), (div, $Q$), $\ldots$ from the action set for that state.

## A.4 Hyperparameters and compute

We used Stable Baselines 3's default hyperparameters for PPO (Table 7), together with the architecture and method constants stated in the body and consolidated in Table 8.

**Compute budget.** All experiments run on a single 24-core CPU machine (no GPU). The PPO `device` setting is left at its Stable-Baselines3 default of `'auto'` (Table 7), but because the machine has no GPU this resolves to CPU for every run reported here. Each `open_large` training is $5 \times 10^6$ steps; the `open_small` headline and seed-matched controls use $3 \times 10^6$, while the beam-width/trigger cohort uses $5 \times 10^6$. With $n_{\mathrm{envs}} = 1$, an `open_large` seed takes 20–30 wall-hours (one CPU core per training; helpers for in-training eval and post-hoc beam-curve eval add up to four more cores transiently). The full $n = 8$ UCB-LP cohort plus $n = 5$ baseline cohort plus the LP-only and intervention-sweep cohorts amount to roughly 1,500–2,000 core-hours of training, dominated by the $5 \times 10^6$-step `open_large` runs. Post-hoc beam-curve evaluations for Figure 1 add an additional $\sim 50$ core-hours. Closed-equation experiments ($10^7$ steps each) account for another $\sim 300$ core-hours across all cohorts. Total: $\sim 2{,}500$ core-hours, no GPU.

| Algorithm | Hyperparameters |
|---|---|
| PPO | $learning\_rate = 0.0003$, $n\_steps = 2048$, $batch\_size = 64$, $n\_epochs = 10$, $gamma = 0.99$, $gae\_lambda = 0.95$, $clip\_range = 0.2$, $ent\_coef = 0.0$, $vf\_coef = 0.5$, $max\_grad\_norm = 0.5$, $normalize\_advantage = True$, $device =' auto'$ |
| PPO-tree | $learning\_rate = 0.0007$, $n\_steps = 5$, $gamma = 0.99$ |

Table 7: Default PPO hyperparameters from Stable Baselines 3. The `device='auto'` default resolves to CPU on our no-GPU machine.

| Constant | Value |
|---|---|
| TreeMLP message-passing rounds $K$ | 3 |
| TreeMLP hidden size $H$ | 128 |
| TreeMLP embedding dim | 64 |
| TreeMLP pooling | mean or max (Sec. 3.1) |
| max state length $L$ | 50 |
| action-set cap $|A|$ | 50 |
| action-diversity penalty $\alpha$ | 0.1 (`open_small` headline); 0 in `open_large` headline[†] |
| value-beam $(\lambda, \beta)$ | $(1, 0)$ (per-class tables; $\lambda = 0$ for headline & ablation) |
| beam width | 5 |
| buffer-refresh period | every 20 rollouts (discard oldest half) |
| CoV per-episode cap | 3 |
| UCB-LP exploration coeff. $\beta$ | 2.0 |

Table 8: Architecture and method constants, consolidated from the body and the training configuration. [†]**Correction (this revision):** the action-diversity penalty is *not* a global constant, as earlier versions implied. It was set to $\alpha = 0.1$ for the `open_small` headline and related penalised cohorts. The `open_large` baseline, LP, UCB-LP and the intervention sweep's non-anti-loop rows, as well as the closed CommonCore runs, used the default $\alpha = 0$; separate anti-loop sweep rows deliberately used $\alpha \in \{0.3, 1.0\}$ (Table 15). This matters for two claims: (i) the seed-level bimodality of Section 4.2.4 is present in the $\alpha = 0$ headline cohorts, so it is *not* an artefact of our reward shaping; (ii) the harm in the matched `open_small` control (Table 11) applies to the cohorts that used $\alpha = 0.1$, including our headline.

## A.5 Closed-equation analysis

This appendix expands the closed-equation results of Section 4.1: a per-structural-type accuracy breakdown, the dominant failure modes, and how performance changes from the small to the large dataset.

*Per-type accuracy.* The "small" dataset contains a heterogeneous mix of equation types. Bucketing the test set by structural class and re-evaluating the trained PPO-TREE-RC-BUF agent reveals a clear pattern (Table 9): linear and radical equations are nearly fully solved ($\geq 0.88$), polynomial equations of degree 2–4 sit in the 0.75–0.82 range, while transcendental forms—particularly trigonometric and logarithmic—are the hardest. The trigonometric bucket (113 equations, 72% accuracy) accounts for most of the remaining

failures. Note that the polynomial buckets contain only "depressed" forms (e.g. $ax^2 + c$, $x^4 + c$) that arise naturally from recursive action application; *true* open polynomials with a linear cross-term ($ax^2 + bx + c$) are not in this dataset and are instead studied in Section 4.2.

Table 9: Greedy test accuracy of PPO-TREE-RC-BUF on the small dataset, broken down by equation class. "poly-deg$k$" includes only the depressed forms reachable by the closed-equation action set (e.g. $ax^2 + c$, $x^4 + c$); equations with a linear cross-term are out of this dataset.

| Type | $n$ | solved | acc |
|------|-----|--------|-----|
| radical (e.g. $\sqrt{x}$) | 47 | 43 | 0.915 |
| linear (poly-deg1) | 110 | 97 | 0.882 |
| quadratic (poly-deg2) | 34 | 28 | 0.824 |
| log | 77 | 58 | 0.753 |
| poly-deg4 | 4 | 3 | 0.750 |
| trig | 113 | 81 | 0.717 |
| **overall** | 387 | 312 | 0.806 |

*Failure modes.* Manual inspection of the failed cases (75 of 387) shows three recurring patterns: (i) trig equations that require an arccos/arcsin step the agent never selects, (ii) log equations whose simplification creates a transcendental that breaks the closed-action set, and (iii) deep polynomials where the agent exceeds the per-episode action budget despite being on a productive path.

*Scaling to the large dataset.* The same per-type evaluation on the "large" dataset (1000 test equations of depth $< 5$) reveals a uniform drop in every category: linear $0.88 \rightarrow 0.60$, quadratic $0.82 \rightarrow 0.40$, log $0.75 \rightarrow 0.27$, trig $0.72 \rightarrow 0.36$, radical $0.92 \rightarrow 0.52$. The drop is largest for deep polynomials (poly-deg4: $0.75 \rightarrow 0.08$). Going from depth $< 4$ to depth $< 5$ approximately quadruples the test-set size and adds substantially deeper nesting, so the harder cases are genuinely harder, not just more numerous.

*Generalization to CommonCore.* On the CommonCore (Poesia) benchmark, PPO-TREE-RC-BUF reaches 0.93 greedy test accuracy on the full benchmark (Table 1), matching ConPoLe's 0.925. The non-buffer PPO-TREE-RC variant scores 0.85 on the full benchmark; restricting the denominator to non-degenerate templates – excluding edge-case equations that lack a free $x$ in the polynomial decomposition, which the closed-action set cannot solve – raises its greedy accuracy to 0.955 (linear 0.98, small rational subset 1.00), and $\geq 0.98$ on the solvable subset. The 0.85 and 0.955 figures for PPO-TREE-RC thus differ only in the denominator (full benchmark vs. non-degenerate subset); both use greedy decoding.

## A.6 Open-equation embeddings (converged vs. stalled seeds)

To diagnose whether the bimodal failure on `open_large` (Section 4.2.4) is a representation problem or a policy problem, we extract the pooled TreeMLP graph embedding for each test equation and project it to two dimensions with PCA and t-SNE. Both the converged seed (`seed109000`; $test_{\text{beam}} = 0.81$) and a stalled seed (`seed108000`; $test_{\text{beam}} = 0.03$) produce clean class clusters (quadratic / cubic / quartic / exponential each form a tight clump in t-SNE space; `abel_level3` is more diffuse). The encoder thus learns the equation-family structure in both cases; the failure is in the policy/value head's ability to act on it, not in the input representation.

## A.7 Change-of-variables grammar ($\pi_{\text{cov}}$)

$\pi_{\text{cov}}$ decodes the substitution as a prefix sequence of grammar productions; production arities make decoding self-delimiting (no explicit length field or brackets needed). For example, $x - b/(2a)$ is emitted as the token sequence SUB X DIV COPY MUL INT2 COPY: SUB takes two children ($x$ and the quotient), DIV takes the copied symbol ($b$) and the product $2a$, MUL takes INT2 and a second copied symbol ($a$), and each COPY names a coefficient symbol of the current equation. Because the model chooses each production – whether to subtract, what to divide by, which coefficients to use – it selects the substitution's structure rather than filling a fixed template.

## A.8 Open-equation solution traces

Below are full greedy solution traces for one held-out test equation from each source class of `open_large`, produced by the converged `seed109` agent. The four change-of-variables classes (quadratic, cubic, quartic, exponential) share the same recipe up to constants (COV → RELABEL → mul(−1) → mul $x$ → collect $x$ → div → RELABEL); only the exponential class requires a *second* CoV at step 4. The depth column counts active CoV substitutions.

| step | action | depth | current lhs (rhs $= 0$) |
|---|---|---|---|
| **quadratic** $- ax^2 + 2cx + 5d = 0$ | | | |
| 1 | COV | 1 | $ax^2 + 5d - c^2/a$ |
| 2 | RELABEL | 1 | $a + bx^2$ |
| 3 | mul(−1) | 1 | $-a - bx^2$ |
| 4 | mul $x$ | 1 | $x(-a - bx^2)$ |
| 5 | collect $x$ | 1 | $x(-a - bx^2)$ |
| 6 | div$(-a - bx^2)$ | 1 | $x$ |
| 7 | RELABEL | 1 | $x\ [= -c/a]$ |
| **cubic** and **quartic**: identical 7-step recipe (only the degree of $x$ differs); see text. | | | |
| **exponential** (nested CoV) $- 2b\,e^x + 2d - 6f\,e^{-x} = 0$ | | | |
| 1 | COV | 1 | $2b/x + 2d - 6fx$ |
| 2 | mul $x$ | 1 | $x(2b/x + 2d - 6fx)$ |
| 3 | EXPAND | 1 | $2b + 2dx - 6fx^2$ |
| 4 | COV (nested) | 2 | $2b + d^2/(6f) - 6fx^2$ |
| 5 | RELABEL | 2 | $a + bx^2$ |
| 6 | mul(−1) | 2 | $-a - bx^2$ |
| 7 | mul $x$ | 2 | $x(-a - bx^2)$ |
| 8 | collect $x$ | 2 | $x(-a - bx^2)$ |
| 9 | div$(-a - bx^2)$ | 2 | $x$ |
| 10 | RELABEL | 2 | $x\ [= -\log(d/f) + \log 6]$ |
| **abel_level3** (closed; no CoV needed) $- ab^2x^2 = 0$ | | | |
| 1 | RELABEL | 0 | $ax^2$ |
| 2 | div $a$ | 0 | $x^2$ |
| 3 | div $x$ | 0 | $x\ [= 0]$ |

Table 10: Per-class greedy solution traces from the converged `open_large` agent (`seed109`, full method stack). The four change-of-variables classes share the same 7-step recipe up to constants; the exponential class is the only one that recruits a *nested* CoV (step 4) after the inner equation becomes a quadratic. The closed-form `abel_level3` example needs no CoV at all and solves in 3 steps.

## A.9 Closed-equation solution traces

Below are the solution traces for each equation in the fixed *closed*-equation environment of Section 4.1. Notice the solution trace is not always optimal. For instance, $ax + b = 0$ could be solved with $(sub, b), (div, a)$, but instead the agent selects $(sub, ax)$ and then $(mul, -1)$. "Truediv" is the SymPy notation for divide.

1. Equation: $a + x = 0$

   ```
   Step 1: a + x = 0 | subtract, a
   Solved: x = -a
   ```

2. Equation: $ax = 0$

   ```
   Step 1: a*x = 0 | div, a
   Solved: x = 0
   ```

3. Equation: $ax + b = 0$

```
Step 1: ax + b = 0 | subtract, ax
Step 2: -ax = b | multiply, -1
Step 3: ax = -b | div, a
Solved: x = -b/a
```

4. Equation: $a/x + b = 0$

```
Step 1: a/x + b = 0 | subtract, b
Step 2: a/x = -b | truediv, 1/x
Step 3: -b*x = a | truediv, b
Step 4: -x = a/b | multiply, -1
Solved: x = -a/b
```

5. Equation: $c(ax + b) + d = 0$

```
Step 1: c*(ax + b) + d = 0 | expand, None
Step 2: acx + bc + d = 0 | subtract, acx
Step 3: -acx = bc + d | multiply, -1
Step 4: acx = -bc - d | truediv, c
Step 5: ax = (-bc - d)/c | truediv, a
Solved: x = (-bc - d)/(ac)
```

6. Equation: $c + d/(ax + b) = 0$

```
Step 1: c + d/(ax + b) = 0 | subtract, c
Step 2: d/(ax + b) = -c | multiply, (ax + b)
Step 3: d = -c (ax + b) | expand, None
Step 4: d = -c ax - c b | multiply, -1
Step 5: -d = c ax + c b | subtract, c b
Step 6: -d - c b = c ax | truediv, c
Step 7: (-d - c b)/c = ax | truediv, a
Step 8: ((-d - c b)/c)/a = x | truediv, 1/a
Solved: x = (-d - c b)/(c a)
```

7. Equation: $cx + d + e(ax + b) = 0$

```
Step 1: cx + d + e(ax + b) = 0 | expand, None
Step 2: aex + be + cx + d = 0 | collect, x
Step 3: be + d + x*(ae + c) = 0 | subtract, x(ae + c)
Step 4: -x(ae + c) = be + d | truediv, ae + c
Step 5: -x = (be + d)/(ae + c) | multiply, -1
Solved: x = -(be + d)/(a*e + c)
```

8. Equation: $e + (ax + b)/(cx + d) = 0$

```
Step 1: e + (ax + b)/(cx + d) = 0 | truediv, 1/(cx + d)
Step 2: (e + (ax + b)/(cx + d))(cx + d) = 0 | expand, None
Step 3: ax + b + cex + de = 0 | collect, x
Step 4: b + de + x*(a + ce) = 0 | subtract, x(a + ce)
Step 5: -x(a + ce) = b + de | expand, None
Step 6: -ax - cex = b + de | collect, x
Step 7: x*(-a - ce) = b + de | truediv, -a - ce
Solved: x = (b + de)/(-a - c*e)
```

---

Algorithm 1: TreeMLP message passing and pooling

---

**Require:** node IDs $\mathbf{X}$, edges $(\text{src}, \text{dst})$, masks $m_{\text{node}}, m_{\text{edge}}$, steps $K$

1: $\text{emb} \leftarrow \text{Proj}(\text{Embed}(\mathbf{X}))$                $\triangleright$ shape $[B, N, H]$

2: $h \leftarrow \text{emb}$

3: **for** $k = 1$ **to** $K$ **do**

4:    $\mathcal{E} \leftarrow \{(i \rightarrow j) \mid m_{\text{edge}}(i \rightarrow j) = 1\}$

5:    $\text{agg}[j] \leftarrow \sum_{(i \rightarrow j) \in \mathcal{E}} h[i]$              $\triangleright$ scatter-sum

6:    $h \leftarrow \text{MLP}\big([\text{emb}, \text{agg}]\big)$

7:    $h \leftarrow h \odot m_{\text{node}}$                $\triangleright$ mask padded nodes

8: **end for**

9: **return** $\text{pool}(h, m_{\text{node}})$                $\triangleright$ mean or max

---

### A.10 TreeMLP Details

**Architecture.** TreeMLP embeds node IDs, projects to a hidden size, and performs $K$ message-passing stages over the (directed) expression tree. At each stage it sums incoming neighbor states, concatenates the result with the fixed input embedding, and applies a two-layer MLP with ReLU. A masked global pool (mean or max) produces the graph embedding exposed to the policy/value heads.

Listing 1: TreeMLP forward pass (minimal).

```
1  def forward(self, obs: dict) -> torch.Tensor:
2      node_ids  = self._to_device(obs["node_features"], self.device)
3      edge_index= self._to_device(obs["edge_index"], self.device)
4      node_mask = self._to_device(obs["node_mask"], self.device)
5      edge_mask = self._to_device(obs["edge_mask"], self.device)
6
7      node_idx = self._id_to_idx(node_ids) if node_ids.dtype != torch.long else
           node_ids
8      B, N = node_idx.shape
9      _, _, E = edge_index.shape
10
11     emb = self.embed(node_idx)                    # [B,N,embed_dim]
12     emb = self.embed_proj(emb)                    # [B,N,H]
13     h = emb.clone()
14
15     src = edge_index[:, 0, :].long()              # [B,E]
16     dst = edge_index[:, 1, :].long()              # [B,E]
17     batch_idx = torch.arange(B, device=self.device).unsqueeze(1).expand(B, E)
18
19     for _ in range(self.K):
20         src_flat = src.reshape(-1)
21         dst_flat = dst.reshape(-1)
22         b_flat   = batch_idx.reshape(-1)
23         m_flat   = edge_mask.reshape(-1).bool()
24
25         src_flat, dst_flat, b_flat = src_flat[m_flat], dst_flat[m_flat], b_flat
               [m_flat]
26         src_idx = b_flat * N + src_flat
27         dst_idx = b_flat * N + dst_flat
28
29         h_flat   = h.reshape(B * N, self.hidden_dim)
30         messages = h_flat[src_idx]                    # [M,H]
31
32         agg_flat = torch.zeros_like(h_flat)           # [B*N,H]
33         agg_flat.index_add_(0, dst_idx, messages)     # sum incoming
34         agg = agg_flat.reshape(B, N, self.hidden_dim) # [B,N,H]
```

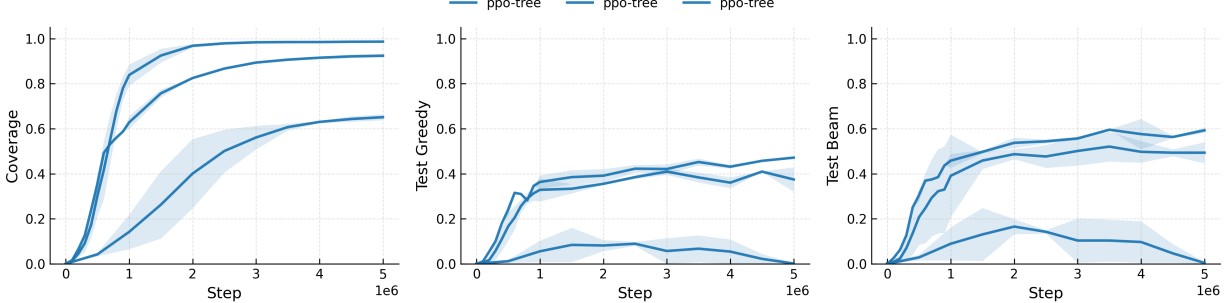

Figure 4: TreeMLP outperforms GCN and GraphSAGE on the small datasets. Mean of three random seeds with shading to min/max.

```
35
36              h = self.node_mlp(torch.cat([emb, agg], dim=-1))    # [B,N,H]
37              h = h * node_mask.unsqueeze(-1).float()
38
39          if self.pooling == "mean":
40              denom = node_mask.sum(dim=1, keepdim=True).clamp(min=1).float()
41              g = h.sum(dim=1) / denom
42          else:
43              if self._minus_inf is None or self._dtype_cache != h.dtype:
44                  self._minus_inf = torch.finfo(h.dtype).min
45                  self._dtype_cache = h.dtype
46              g = h.masked_fill(~node_mask.unsqueeze(-1), self._minus_inf).max(dim=1)
                    .values
47
48          return self.proj(g)
```

**Implementation notes.** We use fixed-size padding with boolean masks for nodes/edges, scatter-add for neighbor aggregation, and pool with mask awareness. All tensors are moved to CUDA when available; no MPS is used.

## A.11  Ablations

We summarize the ablation findings; the corresponding learning curves are in Figures 5 and 4.

- Replacing dense complexity-delta rewards with sparse outcome-only rewards (and disabling the inverse-frequency curriculum) substantially degrades performance. The main drawback of sparse rewards is *far* longer run times: we had to introduce a 1-second-per-step wall-clock timeout to keep training tractable.

- TreeMLP outperforms a vanilla GCN and GraphSAGE on the small dataset in single-seed runs (Figure 4); the tree-aware aggregation appears helpful here, though this comparison is suggestive, not conclusive.

- Curiosity bonuses (ICM, RND, NGU) do not improve TreeMLP learning on `abel_level3` (Figure 6); the body discusses this negative result in Section 3.2. The bonus runs are single-seed and so are suggestive, not conclusive.

## A.12  Deferred main-text figures and tables

This section collects figures and tables deferred from the main text. Each is referenced inline from the appropriate main-text passage.

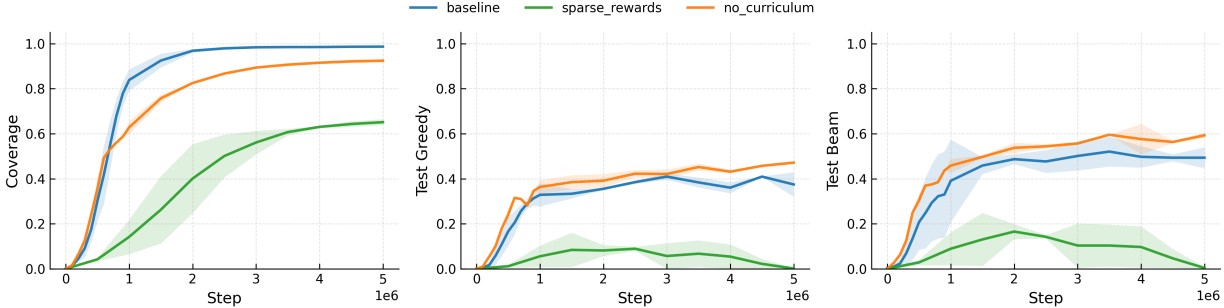

Figure 5: Inductive-bias ablations on the small closed-equation dataset. Each curve removes one component (dense complexity-delta reward + inverse-frequency curriculum, relabel-constants, or the success-replay buffer) from the full PPO-TREE-RC-BUF stack; $y$-axis is greedy test accuracy over training steps (mean of 3 seeds, min/max shading). Conclusion: removing the dense reward/curriculum or the replay buffer substantially degrades learning, so each bias is load-bearing on this dataset.

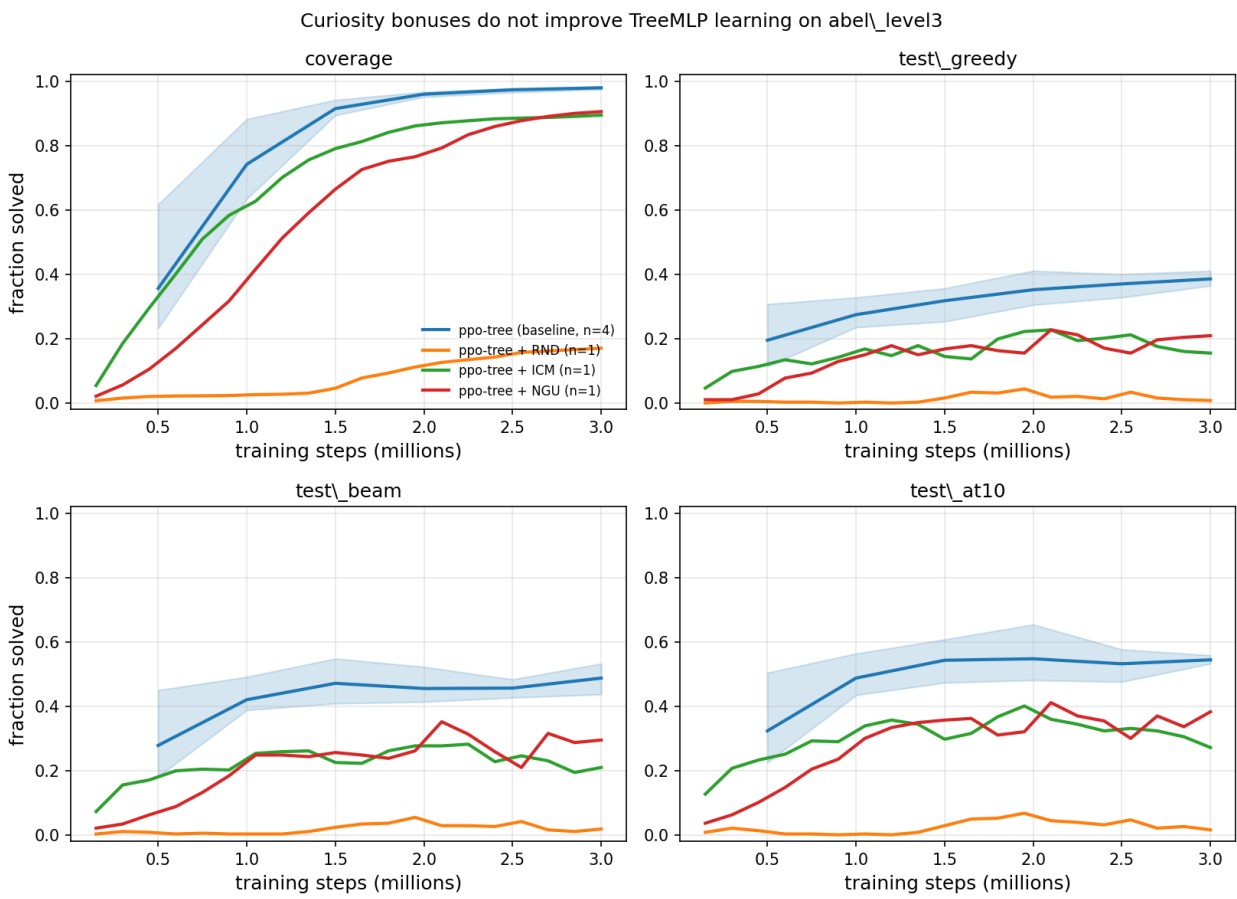

Figure 6: Curiosity bonuses (ICM, NGU, RND) do not improve TreeMLP learning on `abel_level3` (single-seed bonus runs vs. an $n=4$-seed no-curiosity baseline; $y$-axis test metrics over training steps). No bonus matches the baseline; RND fails to learn. These single-seed results are suggestive, not conclusive.

**Cumulative open-equation ablation (referenced from §4.2.2).**

### A.13 CoV trigger ablation: the stalled seeds

The trigger ablation of Section 4.2.3 is run on all eight `open_small` checkpoints of Figure 2, but only the five converged seeds are pooled in Table 4. The three stalled seeds (`147000`, `148000`, `627000`) are reported here and never pooled, for the reason the main text gives: a stalled policy solves $\approx 0$ regardless of who owns the trigger, so these seeds carry no information about trigger quality and would only dilute the comparison.

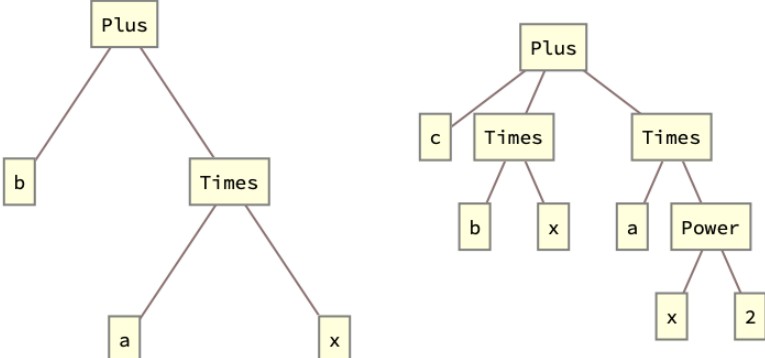

Figure 7: Expression trees for $ax + b$ and $ax^2 + bx + c$. The agent's state vector is a preorder traversal of these trees, padded to length $L = 50$.

**Expression trees (referenced from Sec. 4 state encoding).**

**Curiosity-bonus negative result (referenced from §4.1).**   See Figure 6 and the ablation bullet above; the negative result and its plausible explanation are discussed in Section 3.2.

**TreeMLP embeddings cluster by family (referenced from §4.1).**

**Per-class accuracy on `open_small` (referenced from Section 4.2.2).**

**Per-class accuracy on `open_large` across escape seeds (referenced from Section 4.2.2).**

**Bimodality per-seed dots (referenced from Section 4.2.4).**

**Full $9$-row intervention sweep (referenced from Section 4.2.4).**

**Three competing hypotheses probed (referenced from Section 4.2.4).**   We find no evidence for the three most natural alternative explanations for the bimodal failure; these probes are necessary-not-sufficient:

- *Capacity loss / dormant neurons* Sokar et al. (2023): at the standard ReDo activation threshold ($0.1\times$ per-layer max), *both* stuck and escape seeds have 0% dormant neurons; no evidence of a capacity-loss phenomenon.

- *Data-distribution skew*: an alternative training distribution skewed toward `abel_level3` (the "abel-boost" set) reaches *beam* $= 0.93$ on its matched test split but only 0.04 on the standard test set – the policy overfits to whatever distribution it sees; no evidence that a training-data fix would resolve the bimodality.

- *Encoder representation collapse*: t-SNE projections of the TreeMLP encoder show clean per-class clusters on *stalled* seeds too (Appendix A.6); no evidence of representation collapse.

The evidence is most consistent with a *policy-attractor* phenomenon rather than a capacity, data, or representation failure, though these probes do not exhaustively rule out other explanations.

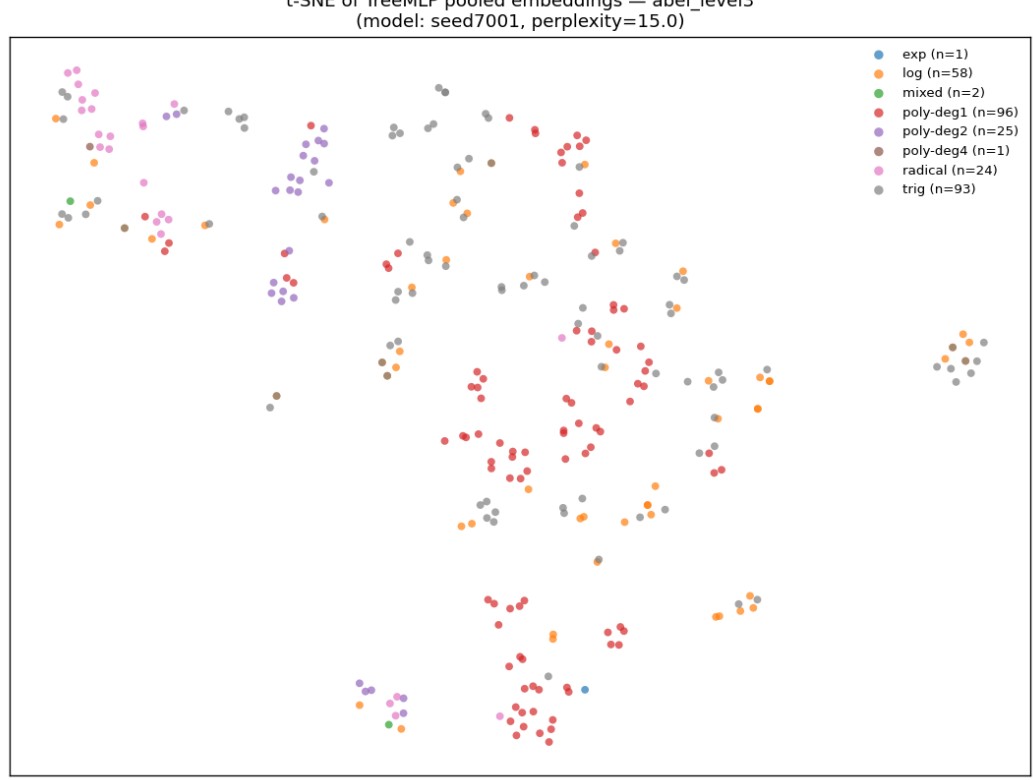

Figure 8: t-SNE projection of TreeMLP pooled embeddings for 300 small-dataset equations, colored by structural type. Clusters correspond to equation family rather than to specific coefficient assignments.

## A.14 Validation-based checkpoint selection protocol

(Referenced from Section 4.2.2.) For every reported standalone-eval beam value, we use a 30/70 validation/test split of the test set with seed 42, evaluate the saved checkpoints on the validation split, select the checkpoint with the highest val beam, and report its test-split beam.

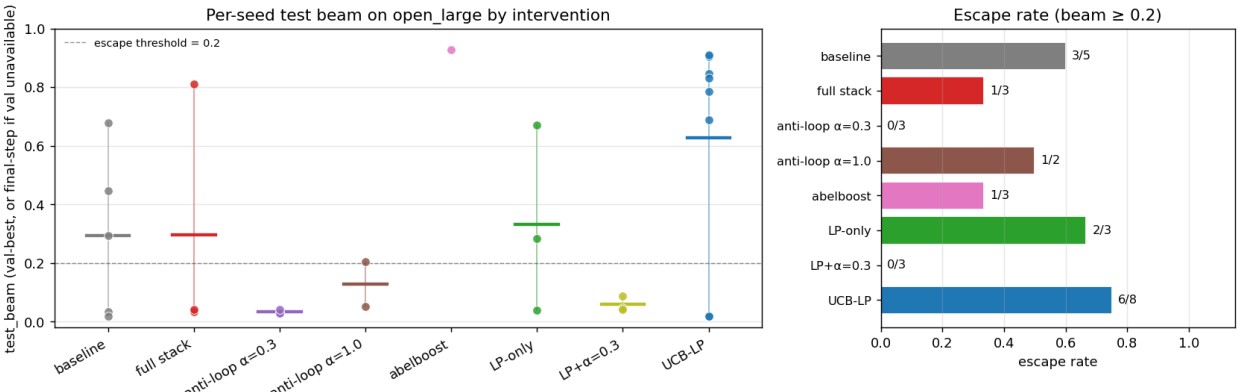

Figure 9: Per-seed beam test accuracy and cohort escape rates from the *initial* 3-*seed screen* on `open_large`. Each dot is a single seed at its validation-best checkpoint; bars show fraction of seeds with $beam \geq 0.2$ (escape threshold). UCB-LP was the only cohort with 3/3 escape in this screen, which motivated expanding it to $n = 8$ (Table 6, where it reaches 6/8). Numbers in Table 6 supersede this screen for the three headline cohorts.

| configuration | $test_\text{beam}$ per seed | | | mean $\pm$ sd |
|---|---|---|---|---|
| | 507000 | 508000 | 509000 | |
| C1 baseline (PPO-TREE-RC-BUF-COV) | 0.250 | 0.406 | 0.406 | $0.354 \pm 0.090$ |
| C2 + action-diversity (anti-loop) penalty | 0.109 | 0.109 | 0.109 | $0.109 \pm 0.000$ |
| C4 + cube-root & nested CoV (flat buffer) | 0.484 | 0.766 | 0.859 | $\mathbf{0.703 \pm 0.195}$ |
| C5 + freshness-managed buffer (**full stack**) | 0.844 | 0.094 | 0.531 | $0.490 \pm 0.377$ |
| *Paired within-seed increments* (per-seed values; mean $\pm$ sd over the three shared seeds): | | | | |
| C1 $\rightarrow$ C2 action-diversity penalty | $-0.141, -0.297, -0.297$ | | | $\mathbf{-0.245 \pm 0.090}$ |
| C2 $\rightarrow$ C4 cube-root & nested CoV | $+0.375, +0.656, +0.750$ | | | $\mathbf{+0.594 \pm 0.195}$ |
| C4 $\rightarrow$ C5 freshness-managed buffer | $+0.359, -0.672, -0.328$ | | | $-0.214 \pm 0.525$ |
| *End-to-end* (baseline $\rightarrow$ each endpoint; the contrast the ladder's adjacent rungs do not show): | | | | |
| C1 $\rightarrow$ C4 (no freshness buffer) | $+0.234, +0.359, +0.453$ | | | $\mathbf{+0.349 \pm 0.110}$ |
| C1 $\rightarrow$ C5 (**full stack**) | $+0.594, -0.312, +0.125$ | | | $+0.135 \pm 0.453$ |
| *Removing the penalty* ($\alpha = 0$, same seeds and budget; new in this revision): | | | | |
| C6 = C4 at $\alpha = 0$ | 0.859 | 0.906 | 0.859 | $\mathbf{0.875 \pm 0.027}$ |
| C7$^\ddagger$ = C5 at $\alpha = 0$ (full stack) | 0.875 | 0.875 | 0.922 | $\mathbf{0.891 \pm 0.027}$ |
| C4 $\rightarrow$ C6 (drop $\alpha$) | $+0.375, +0.141, +0.000$ | | | $+0.172 \pm 0.189$ |
| C5 $\rightarrow$ C7 (drop $\alpha$; full stack) | $+0.031, +0.781, +0.391$ | | | $\mathbf{+0.401}$ (3/3) |
| C1 $\rightarrow$ C6 (clean end-to-end, no freshness) | $+0.609, +0.500, +0.453$ | | | $\mathbf{+0.521 \pm 0.080}$ |
| C1 $\rightarrow$ C7 (**clean end-to-end, full stack**) | $+0.625, +0.469, +0.516$ | | | $\mathbf{+0.537 \pm 0.080}$ |

Table 11: **Seed-matched open-equation ablation** on `open_small` ($test_\text{beam}$, beam width 5, $\lambda = 0$, the 64-equation held-out slice; every cell trained 3M steps). New in this revision at a reviewer's request, replacing v5's single-seed cumulative table, whose rows came from different seeds, reported a training peak (0.341) rather than a final value (0.275) for the action-diversity row, and differed in `max_steps` — so none of its increments could be read causally. Here all configurations share the same three seeds, the same budget and the same decoder, with no early stopping and no checkpoint selection, so no configuration can win by being frozen at a luckier step. (v5's "+ value-guided beam" row was a *decode-time* change, not a training ingredient; we hold the decoder fixed instead of making it a rung.) **Findings.** The change-of-variables machinery carries the result (+0.594, positive on every seed; it bundles the cube-root primitive with nested CoV, which this design cannot separate). The auxiliary RL engineering does not: the action-diversity penalty is *harmful* ($-0.245$, negative on every seed, against v5's reported $+0.03$), and the freshness-managed buffer is unresolvable (its increment changes sign across seeds). End-to-end, the lift over baseline is carried by the simpler C4 (+0.349, positive on every seed); the penalised full stack's +0.135 flips sign and is *not* resolvable at $n = 3$. **What this licenses.** Only large effects. Re-running an identical (configuration, seed) cell reproduces $test_\text{beam}$ to within 0.079 on average (0.253 worst case). That estimate is itself limited: it comes from $n = 6$ replicate cells on C1/C2 — the two lowest-variance configurations — at 1M steps on the full 91-equation file, so it is extrapolated across budget and denominator to this table and is a *floor*, not a bound; a *difference* of two cells carries $\sim\sqrt{2}$ that noise. We therefore claim only increments whose *sign* is stable across all three seeds, never point estimates; 3/3 sign agreement carries a floor two-sided $p$ of 0.25, so these are corroborated directions, not powered tests. This is an ablation of credit among ingredients and is *not* a re-measurement of the headline, which rests on a larger 5-seed cohort at a validation-selected checkpoint (Table 2). One cell (C5/`seed509000`) is a re-run: its original trial died in a SymPy `NaN`-comparison fault after 96 s and was retrained from scratch under the identical training configuration at the same seed (509000), though alone on the machine rather than alongside its cohort. $^\ddagger$C7/`seed507000`'s original trial died at $\sim$2.8M steps to the SymPy/mpmath overflow on large rational coefficients (also seen for two `open_large` seeds; Section 6) and was retrained from scratch at the same seed to complete the row. **At** $\alpha = 0$, both C6 and C7 exceed the reported headline within this fixed-step control, variance collapses, and none of the six C6/C7 cells stalls (including C1, 0/9 $\alpha = 0$ cells stall). We do not promote either control mean to the headline because the downstream `open_small` analyses and validation-selected headline use a different, $\alpha = 0.1$ cohort; nor do we call 0.79 a lower bound (Section 6).

Table 12: **Trigger ablation on the three stalled `open_small` seeds** (greedy, $n = 91$). Every trigger regime — including **never** — lands within *one equation* ($1/91 = 0.011$) of every other on each seed. The trigger is irrelevant when the policy has not learned the closed-action steps needed to finish an episode after a CoV, which is precisely why these seeds are excluded from Table 4 rather than averaged into it.

| seed | never | always@0 | rule | learned | rule + expand |
|---|---|---|---|---|---|
| 147000 | 0.000 | 0.000 | 0.000 | 0.000 | 0.000 |
| 148000 | 0.011 | 0.011 | 0.011 | 0.011 | 0.011 |
| 627000 | 0.110 | 0.110 | 0.110 | 0.099 | 0.110 |

Table 13: Per-class test accuracy on `open_small` (value-guided beam). Baseline is the single-seed PPO-TREE-RC-BUF-COV reference ($n = 1$); full-stack is the 4-seed mean of seeds completing the full budget (the 5th early-stopped seed is excluded here, whereas the headline 0.79 in Section 4.2.2 is the 5-seed mean that includes it – the two are therefore not directly comparable). The $n$-weighted overall of these 4-seed per-class values is $\approx 0.85$ on the full 91 equations; the headline 0.79 is on the 64-equation test slice.

| class ($n$) | baseline | full stack |
|---|---|---|
| quadratic (15) | 0.00 | 0.97 |
| cubic (15) | 0.00 | 1.00 |
| quartic (15) | 0.00 | 1.00 |
| exponential (15) | 0.00 | 0.88 |
| abel_level1 (1) | 1.00 | 1.00 |
| abel_level2 (10) | 0.50 | 0.75 |
| abel_level3 (20) | 0.25 | 0.53 |

Table 14: Per-class `open_large` test accuracy (value-guided beam, 200 eqns/class) across the 3 escape seeds (*overall* $\in [0.79, 0.91]$); the 6 stall seeds are $\approx 0$ on every class. The four CoV classes saturate while `abel_level3` remains weak and high-variance – the open_small pattern, sharpened at scale.

| class ($n = 200$) | escape-seed mean [min, max] |
|---|---|
| quadratic | 0.98 [0.98, 0.99] |
| cubic | 0.99 [0.98, 1.00] |
| quartic | 0.99 [0.98, 1.00] |
| exponential | 0.98 [0.96, 1.00] |
| abel_level3 | 0.39 [0.07, 0.55] |

Table 15: **Initial 3-seed screen** of the full intervention sweep on `open_large` (mean validation-best beam over 3 seeds; escape = *beam* $\geq 0.2$). All cohorts were first screened at 3 seeds to select interventions; the three headline cohorts were subsequently expanded to $n = 5$ (baseline), 3 (vanilla LP), and 8 (UCB-LP) seeds, with final numbers in Table 6, which *supersede* the corresponding rows here. The non-headline rows (anti-loop, dataset rebalance, LP+anti-loop, LP++) were screened only and not expanded.

| Cohort | Mean beam | Escape rate |
|---|---|---|
| Baseline (full stack)[*] | 0.34 | 2/3 |
| Anti-loop ($\alpha = 0.3$) | 0.05 | 0/3 |
| Anti-loop ($\alpha = 1.0$) | 0.11 | 1/3[†] |
| Dataset rebalance (abelboost) | 0.04 | 0/3 |
| LP curriculum (vanilla)[*] Matiisen et al. (2019) | 0.40 | 2/3 |
| LP + anti-loop ($\alpha = 0.3$) | 0.07 | 0/3 |
| LP++ (optim. init + EMA + boot) | 0.10 | 0/3 |
| **UCB-LP curriculum**[*] | **0.81** | **3/3** |
| Oracle (SymPy) | 1.00 | – |

[*] headline cohort; expanded beyond this 3-seed screen, see Table 6 for the superseding numbers. [†] one of three seeds crashed mid-run due to a SymPy/mpmath `OverflowError` on a large rational; we report rate over surviving seeds.

