# OpenReview forum: "Reinforcement Learning for Symbolic Equation Solving"
_TMLR — Under review for TMLR_

### Review · Reviewer_oNqP · 2026-07-05

**Summary Of Contributions:**

This paper studies reinforcement learning for step-by-step symbolic equation solving. The goal is not merely to output a final answer, but to learn a policy that produces a sequence of algebraic transformations leading to a solution. This is a practically meaningful problem: many downstream applications, such as tutoring systems, theorem-prover tactics, and symbolic manipulation pipelines, require an explicit and checkable procedure rather than a one-shot answer.

The paper formulates symbolic equation solving as a Markov decision process with a dynamic, per-state action space. At each state, the available actions are constructed from algebraic operations paired with sub-expressions of the current equation. The authors propose TreeMLP, a tree-structured policy/value network for encoding symbolic expression trees, and train it with PPO. The full system further incorporates several domain-specific mechanisms, including curriculum sampling, constant relabeling, success replay, action-diversity penalties, value-guided beam search, and macroactions such as expand, collect, and change of variables.

A major contribution is the extension from closed equations to restricted open-equation families. In this setting, the agent must invoke a change-of-variables operation to introduce structure that is not present in the original equation. The paper handles this by exposing CoV as a macroaction. The main RL policy learns when to trigger this macroaction and how to combine it with ordinary algebraic operations. The substitution generator itself is trained separately, but the authors show through a pipeline decomposition that replacing it with a CAS-based substitution has little effect on end-task accuracy, indicating that the learned trigger and subsequent symbolic-action policy are the main contributors to the reported open-equation performance.

The empirical results are strong within the proposed benchmark setting. On closed CommonCore equations, the full TreeMLP-PPO system achieves 0.93 greedy test accuracy, matching the previously reported ConPoLe result of 0.925. On the open_small benchmark, the full method reaches 0.79 beam-decoded test accuracy, compared with 0.31 for the plain open-equation baseline. The paper also reports an important scaling phenomenon on open_large: runs with identical code, data, and hyperparameters exhibit a clear seed-level bimodality, either escaping to a high-performing regime or collapsing into a class-level policy trap. The analysis of this failure mode, together with the UCB-style learning-progress curriculum proposed to mitigate it, is one of the more interesting empirical parts of the paper.

The main strengths of the paper are the importance and relative underexploration of the problem, the natural dynamic-action MDP formulation, the use of a tree-structured policy network for symbolic states, the integration of higher-level algebraic macroactions, and the production of verifiable solution traces. The paper is also relatively transparent about several limitations, including the reliance on beam decoding in the open setting, the restricted nature of the open-equation families, the supervised CoV substitution generator, and the seed-level instability at scale.

The main weaknesses are also clear. The comparison to prior work is limited, especially for the open-equation setting where the strongest baselines are mostly internal or search-based rather than learned systems of comparable scope. Several experimental conclusions rely on small numbers of seeds or cumulative ablations that are not fully controlled. The open-equation results are much stronger under beam search than under greedy decoding, so the final performance reflects policy-guided search rather than a consistently reliable top-1 policy. Finally, the benchmark is still substantially shaped by the hand-designed action set and the chosen CoV templates, which limits the extent to which the results demonstrate general symbolic equation-solving ability.

**Audience:**

Yes

**Audience Explanation:**

This paper should be of interest to several parts of the TMLR audience, especially researchers working on reinforcement learning, neural-symbolic reasoning, procedural mathematical reasoning, learned search, and symbolic manipulation.

**Broader Impact Concerns:**

I do not see major broader-impact concerns. The equations considered in this paper are already solvable by standard computer algebra systems, and the method does not appear to introduce a new harmful capability. The authors' existing broader impact statement is mostly adequate.

**Claims And Evidence:**

Yes

**Claims Explanation:**

The evidence is convincing for the paper's core technical claims. The closed-equation experiments show that the proposed TreeMLP-PPO system, together with constant relabeling and success replay, substantially outperforms a vanilla PPO baseline and reaches 0.93 greedy accuracy on the CommonCore benchmark, matching the reported ConPoLe result of 0.925. The open-equation experiments also support the usefulness of exposing change of variables as a macroaction: on open_small, the full method reaches 0.79 beam-decoded test accuracy compared with 0.31 for the plain open-equation baseline. The pipeline decomposition is particularly helpful: masking CoV nearly eliminates performance, while replacing the learned CoV substitution generator with a SymPy-based substitution changes end-task accuracy only slightly. This supports the authors' claim that the main learned component is the CoV trigger and the subsequent symbolic-action policy, not the substitution generator itself.

The evidence for the open large seed-level bimodality is also credible. The per-seed results show a clear separation between escape and stall regimes, and the class-level analysis gives a plausible explanation in terms of policy trapping. The UCB-LP curriculum results are promising, and the authors appropriately acknowledge that the escape-rate improvement is not statistically powered.

The main issue is that some headline statements are broader than the evidence warrants. "From reward alone" should be restricted to the main RL policy and the absence of supervised solution traces, since the CoV substitution generator is supervised. "Open equations" should be consistently qualified as a restricted-open benchmark with hand-designed families and known CoV structures. The paper should also state more clearly whether success means finding one valid root or producing the complete solution set. Finally, the strongest open-equation result is a beam-search result, while greedy accuracy is much lower; the claim should therefore be framed as policy-guided search rather than reliable top-1 procedural solving.

Overall, the evidence supports the paper's main contribution: a learned policy can guide step-by-step symbolic manipulation and learn when to invoke a CoV macroaction in the proposed restricted benchmark. The evidence does not support a broad reading in which the method is a general reward-only solver for open symbolic equations. The paper should revise its high-level wording so that the abstract, introduction, and conclusion match the actual empirical scope.

**Requested Changes:**

1. Tighten the high-level claims and terminology.

The abstract, introduction, and conclusion should be revised so that the paper's scope is explicit. The phrase "from reward alone" should be used only for the main RL policy and the absence of supervised solution traces; the CoV substitution generator is supervised. Similarly, "open equations" should be consistently described as restricted open-equation families with known CoV forms, not as general open symbolic equation solving. The paper should also avoid using "solves equations" in a way that implies complete symbolic solving when the evaluation protocol rewards solving under the benchmark-specific criterion.

2. Define the mathematical semantics of solving and trace validity.

The paper should state clearly whether the target is to find one satisfying root, all roots, or an equivalence-preserving derivation. This distinction matters because the paper counts multi-root cases as solved when any one root is found. The authors should also specify the algebraic domain assumptions, such as real vs. complex variables, and explain how square roots, logarithms, trigonometric and inverse-trigonometric operations, branch choices, extraneous roots, and lost roots are handled. Since the paper emphasizes verifiable traces, it should make explicit whether every intermediate step is checked for equivalence or whether only the final candidate solution is verified by substitution into the original equation.

3. Make the empirical evidence easier to interpret.

The main result presentation should more clearly separate greedy accuracy, beam-decoded accuracy, beam width, validation-based checkpoint selection, and search budget or inference cost. This is especially important for the open-equation experiments, where the gap between greedy and beam-decoded performance is large. The ablation results should also be presented cautiously: several rows are cumulative, single-seed, or not directly comparable, so they are useful diagnostics but should not be read as precise component-wise effect sizes. Similarly, the UCB-LP curriculum should remain framed as an exploratory mitigation of the open_large bimodality rather than a statistically established fix.

4. Improve comparison and reproducibility.

The paper should position its baselines more clearly. Although directly comparable learned baselines for nonlinear and restricted-open symbolic equation solving are scarce, the current open-setting comparison is mostly internal or search-based. The authors should strengthen the discussion of BFS, best-first/A* search, CAS oracle results, and the limited comparability of ConPoLe outside the closed-equation setting. The work would also be stronger with an anonymous code/data release during review, or at minimum a detailed reproducibility package covering dataset generation, action masks, CoV grammar, random seeds, validation/test splits, and evaluation scripts.

---

> ### Author Response · Authors · 2026-07-08
> **Reply to reviewer oNqP**
>
> We thank the reviewer for a careful report and agree with every requested change; the revision narrows the language to the empirical scope and adds the missing formal semantics.
>
> **RC1 — Tighten claims/terminology.** Revised throughout the abstract, introduction, contributions, and conclusion: "from reward alone" is now scoped to the main RL policy (naming the supervised CoV generator); "open" is consistently "restricted-open" (hand-designed families with known CoV forms), with an explicit "we do not claim general open-equation solving"; and success is stated up front as producing *one* root verified by substitution, not the complete solution set.
>
> **RC2 — Semantics of solving and trace validity.** A new paragraph in Section 3 states, in one place: (i) an episode is solved iff the tracked lhs is the solve variable, the rhs is free of it, and the candidate root verifies to zero on substitution under op-count-guarded canonicalizers — one root, not the solution set; (ii) coefficients are real, sqrt/log/inverse-trig use principal branches; (iii) extraneous roots are eliminated by the terminal check, and lost roots are handled by masking divisions that vanish at a root plus the one-root criterion; (iv) intermediate steps are exact and replayable but not individually certified equivalent — correctness is enforced at termination, with CoV verification against the tracked equation and unwinding by exact inverse composition. All verification failures are conservatively counted as unsolved.
>
> **RC3 — Interpretability of the evidence.** A "Reporting conventions" paragraph (Section 4.2.2) separates and labels greedy vs. beam, states the beam width, validation-based checkpoint selection (30/70 split), and per-equation inference cost. The single-seed cumulative ablation is framed as diagnostics (not effect sizes) and, per Reviewer Fmhx, moved to the appendix. UCB-LP is reported with the Fisher/Wilson statistics and described uniformly as a candidate mitigation showing a non-significant positive trend. (See also the General Response: we corrected a stale greedy number, 0.24 → 0.67; the new Table 2 shows the greedy policy alone exceeds the non-learned search baselines.)
>
> **RC4 — Comparison and reproducibility.** The baseline-positioning paragraph (Section 4.2.2) now gives the A* budget and heuristic, explains why no learned external baseline exists for restricted-open (ConPoLe's axioms contain no substitution-introducing operation; the stack-calculator interface of Dabelow & Ueda does not transfer), and cites the SymPy oracle as the 1.00 upper bound — framed as a limitation. The Reproducibility Statement commits to an anonymized repository during the discussion period (environment + terminal-verification code, dataset/split generation, the pi_cov grammar, all seeds and launch configs, and the greedy/beam/BFS/A* evaluation scripts).
>
> We agree with the reviewer's summary that the evidence does not support a general reward-only open solver; the revised claims are exactly (i) main-policy-only reward learning, (ii) restricted-open families, (iii) one-verified-root success, and (iv

---

### Review · Reviewer_Fmhx · 2026-07-08

**Summary Of Contributions:**

This paper presents the first RL agent for step-by-step solving of **open** (change-of-variables) symbolic equations from reward alone, without supervised solution traces. Prior RL for equation solving is confined to linear equations (ConPoLe; Dabelow & Ueda 2024); this work extends the scope to nonlinear and open families (quadratic, cubic, quartic, exponential).

**Key contributions:**
1. An MDP formulation with a dynamic per-state action space (operation × sub-expression), paired with TreeMLP — a tree-structured policy/value network using sum-aggregation message passing without degree normalization — as the PPO backbone.
2. A change-of-variables (CoV) macroaction: the policy learns *when* to trigger a CoV; the substitution itself comes from a supervised generator $\pi_\text{cov}$ interchangeable with a CAS ($\leq 0.02$ end-task effect), decoupling trigger learning from substitution generation.
3. On `open_small`: 5-seed beam-decoded test accuracy 0.79 (vs. 0.31 baseline); on closed CommonCore: 0.93 (matching ConPoLe's 0.925). Greedy accuracy is only $\approx$0.24, so performance is dominated by decode-time search.
4. Characterization of a seed-level bimodality at 10$\times$ scale (escape vs. stall regimes with no seed in the gap), with a UCB-LP curriculum showing a non-significant positive trend (6/8 escape vs. 3/5 baseline; Fisher $p \approx 1.0$).
5. Transparent disclosure of limitations: the beam/greedy gap, the statistically underpowered UCB-LP result, and the restricted scope of "open" equations.

**Key strengths:**
- Internally consistent method design: TreeMLP's architecture choices align well with expression-tree structure; each inductive bias (curriculum, success replay, constant relabeling) has separable contributions documented in Table 1.
- The $\pi_\text{cov}$–CAS decomposition (Table 3) compellingly supports the "trigger, not substitution" framing.
- The bimodality analysis (Section 4.2.3) is rigorous: three alternative hypotheses are systematically eliminated, with an appropriately scoped "necessary-not-sufficient" epistemic qualification.
- Experimental transparency follows best practice (per-seed reporting, 30/70 val/test split, explicit crash-seed counting).

**Key weaknesses:**
- The concurrent work O'Keeffe (2025, NeurIPS MATH-AI Workshop, arXiv:2510.17022) already solves nonlinear closed equations with PPO + graph actions; the "first nonlinear" claim needs to be narrowed to "first open."
- The headline result (beam=0.79) relies on decode-time search; greedy$\approx$0.24 means the policy's top-1 action is wrong in most states. Non-learned search (BFS 0.26, A* 0.38–0.64) also substantially exceeds greedy, making it unclear how much of the 0.79 comes from the learned policy vs. the search procedure.
- UCB-LP is presented as Contribution (c) despite Fisher $p \approx 1.0$; this is an overstatement.

**Audience:**

Yes

**Audience Explanation:**

The paper's core question is relevant to the communities that TMLR's audience is interested in, such as RL methodology, symbolic reasoning, and formal mathematics.

**Broader Impact Concerns:**

No concerns.

**Claims And Evidence:**

Yes

**Claims Explanation:**

The paper's core claims are generally supported by adequate evidence, with important caveats on two points:

**Well-supported:**
- CommonCore accuracy (0.93 vs. ConPoLe 0.925) is directly comparable under greedy decoding across 3 seeds, with a full ablation in Table 1.
- $\pi_\text{cov}$ independence: swapping with SymPy's exact depression moves accuracy by $\leq 0.02$ across four escape seeds (Table 3), strongly supporting the "trigger, not substitution" claim.
- Bimodality existence: among 16 training runs, results cleanly partition into escape ($\geq 0.2$) and stall ($\leq 0.10$) regimes with no seed in the empirical gap, a robust classification. The three diagnostic probes (dormant neurons, data distribution, encoder collapse) are properly scoped as necessary-not-sufficient.
- The per-class trap analysis provides a mechanistic explanation consistent with the escape/stall pattern.

**Some Weaknesses:**
- The main result beam=0.79 depends on width-5 beam search at test time; greedy mean is only $\approx$0.24. The abstract's "solves equations" phrasing overstates what the policy alone achieves. Non-learned search baselines (BFS 0.26, A* 0.38–0.64) show that a substantial fraction of the beam-5 performance comes from search, not the policy. The paper is transparent about this (Section 4.2.2), but the headline framing could mislead uninformed readers.
- The UCB-LP escape-rate change (3/5$\to$6/8) has Fisher exact $p \approx 1.0$, meaning no statistical evidence of improvement. The authors correctly call it "promising, not demonstrated," but listing it as Contribution (c) and repeatedly using "partially mitigate" overweights the finding relative to its evidence base.

**Requested Changes:**

**1. Correct the "first nonlinear" priority claim relative to O'Keeffe (2025).**
O'Keeffe (2025, NeurIPS 2025 MATH-AI Workshop, arXiv:2510.17022) already demonstrates PPO + graph-structured action spaces solving nonlinear closed equations (radicals, exponentials, trig). The paper cites this work but does not adjust its "first" claims. Specific changes needed:
- Replace "first RL agent for nonlinear symbolic equations" in the abstract, introduction, and contributions with "first RL agent for **open** symbolic equations" or equivalent.
- Add a dedicated comparison paragraph in Section 2 detailing the technical differences: (a) O'Keeffe is limited to closed equations (no CoV); (b) O'Keeffe relies on curiosity-based exploration, which this paper shows fails under TreeMLP; (c) this work adds success replay, constant relabeling, beam decoding, and the bimodality analysis.

**2. Clarify the scientific interpretation of beam-search results.**
The 3.3$\times$ gap between beam (0.79) and greedy (0.24) means the policy's top-1 action is unreliable. Non-learned search (BFS 0.26, A* 0.38–0.64) also far exceeds greedy. Required:
- A direct three-way comparison (greedy vs. beam vs. non-learned search) quantifying how much of the 0.79 comes from the policy vs. the search procedure.
- Explicit discussion of whether, without beam search, the method outperforms O'Keeffe (2025)'s test\_10 $\approx$ 0.8–1.0 (which uses stochastic rollouts, not beam).
- Ideally, a multi-seed beam-width ablation on `open_small`.

**3. Tone down the UCB-LP contribution claim.**
Fisher exact $p \approx 1.0$ provides no statistical evidence of improvement. Revise Contribution (c) to describe UCB-LP as "a candidate mitigation direction showing a non-significant positive trend" rather than a demonstrated partial mitigation.

**4. Move single-seed ablation rows (Table 2, rows 1–4) to the appendix.**
These are labeled "illustrative" but appear in the main paper in a setting with documented seed-level variance exceeding 0.7. They are not interpretable as effect sizes and risk misleading readers.

---

> ### Author Response · Authors · 2026-07-08
> **Reply to reviewer Fmhx**
>
> We thank the reviewer for a precise report and for recognizing the method design, the π_cov–CAS decomposition, and the bimodality analysis. We agree with all four requested changes.
>
> ### RC1 — Correct the "first nonlinear" priority claim relative to O'Keeffe (2025)
>
> **Conceded and narrowed throughout.** Every "first" now attaches to the **open / change-of-variables** setting. The abstract, introduction, contribution (a), and conclusion no longer claim "first nonlinear"; they credit O'Keeffe (2025) with prior RL for nonlinear *closed* equations and state that our first-of-kind contribution is the restricted-open (CoV) setting. We also corrected the adjacent over-statement that "prior RL is limited to linear equations." A **new Section 2 comparison paragraph** covers the three differences the reviewer requested: (a) O'Keeffe (2025) is closed-only (no substitution-introducing action); (b) it relies on curiosity bonuses (ICM/RND/NGU), which we find *fail* to match a no-curiosity baseline under TreeMLP; (c) we add TreeMLP, success replay, constant relabeling, value-guided beam, and the bimodality analysis.
>
> ### RC2 — Clarify the scientific interpretation of beam-search results
>
> **In building the requested three-way comparison we found and corrected the stale greedy number (≈0.24 → 0.67; see the General Response), which resolves the concern in the strongest way.**
>
> - **Three-way comparison (new Table 2).** The learned policy's greedy top-1 rollout (0.67) alone exceeds the strongest non-learned search over the same action space (A* up to 0.64, BFS 0.26); value-guided beam adds a further ≈0.12 to 0.79 at ~7× less per-equation search. The accuracy is policy-dominated: search contributes a margin, not the bulk of the result. The SymPy oracle upper-bounds the benchmark at 1.00.
> - **Comparison to O'Keeffe (2025)'s decoder.** That work reports test_10 ∈ [0.8, 1.0] under stochastic rollouts. We added the matching decoder — best of 10 sampled trajectories — and obtain **success@10 ≈ 0.90** on the strong open_small seeds, competitive despite the harder open setting (their number is on closed equations).
> - **Multi-seed beam-width ablation (new Figure 2).** Value-guided-beam accuracy vs. decode width (width 1 = greedy) across eight full-stack open_small checkpoints; on the five converged seeds the mean rises from greedy (0.73) through width 5 (0.84) and then saturates, supporting width 5 as the operating point. Three seeds stalled (greedy ≈0 at every width — consistent with the seed-level bimodality, and uninformative about width).
>
> ### RC3 — Tone down the UCB-LP contribution claim
>
> **Adopted the reviewer's language.** Contribution (c) now reads "a candidate mitigation direction showing a non-significant positive trend (3/5 → 6/8 escape, two-sided Fisher exact p ≈ 1.0), not a demonstrated fix." Every "partially mitigate"/"partially reduce" occurrence has been downgraded uniformly (abstract, introduction, conclusion, and the residual-variance paragraph of Section 4.2.3).
>
> ### RC4 — Move single-seed ablation rows to the appendix
>
> **Done.** The cumulative single-seed configuration table is now in the appendix (Table 9). The main text keeps only the endpoints (0.31 baseline → 0.62 single-seed → 0.79 five-seed full stack) in prose, with the caveat that every non-final row is a single-seed diagnostic that should not be read as an effect size, and a pointer to the appendix table.

---

### Review · Reviewer_BV6L · 2026-07-11

**Summary Of Contributions:**

The paper models step by step algebra equation solving as a reinforcement learning problem with a dynamic action space. It trains a small tree based GNN policy using PPO. The main contribution is solving restricted open equations, which require a change of variables (CoV), such as completing the square, instead of only rearranging existing terms. The CoV itself is generated by a small supervised model, while RL learns when to apply it and how to combine it with other operations. The method achieves strong results on four CoV equation families, matches ConPoLe on the CommonCore benchmark, and reports that larger models often either learn the task well or fail to learn CoV at all.

**Audience:**

Yes

**Audience Explanation:**

The paper addresses reinforcement learning for symbolic reasoning, procedural problem solving, and neural symbolic learning, all of which are relevant topics for the TMLR community. Although the benchmark is specialized, the formulation of symbolic equation solving as an MDP with a dynamic action space, together with the analysis of reinforcement learning behavior and training dynamics, provides useful insights that extend beyond this particular application.

**Broader Impact Concerns:**

I do not have significant broader impact concerns

**Claims And Evidence:**

Yes

**Claims Explanation:**

The main claims are generally supported by the presented evidence. The paper evaluates the proposed method on both closed and restricted open equation benchmarks, compares against search based baselines, provides ablation studies, analyzes training across multiple random seeds, and clearly discusses the limitations of the proposed approach. The authors are appropriately cautious when interpreting the UCB curriculum results and explicitly state that they represent only a positive trend rather than a statistically significant improvement. However, one central claim remains only indirectly supported. The paper argues that the policy learns when to invoke the change of variables, but this is not evaluated directly through trigger specific metrics or simple trigger baselines.

**Requested Changes:**

Critical
Add an experiment that directly evaluates the learned change of variables trigger, for example by comparing against a simple rule based trigger or reporting trigger accuracy.

Would strengthen the paper
Repeat the main open setting ablation experiments with multiple random seeds, since the paper itself demonstrates substantial seed dependent variability.

---

> ### Author Response · Authors · 2026-07-20
> **Reply to reviewer BV6L**
>
> We thank the reviewer for identifying precisely the one central claim our evidence did not directly support. We ran the requested experiment. It changed our conclusion, and we revised the paper accordingly rather than defending the original wording.
>
> RC1 (Critical) — Directly evaluate the learned CoV trigger
> Done, and it narrowed the claim. New Section 4.2.3, "Is the CoV trigger learned, or would a rule do?" (Tables 4 and 5; stalled seeds in Appendix A.13). We hold the trained policy fixed, mask the CoV action, and hand the trigger to an external rule; the policy still selects every non-CoV action. Four hand-coded regimes plus the learned policy, all greedy-decoded on the same checkpoints.
>
> Two things are true and we state both.
>
> (i) On three of four families, nothing is learned. For quadratic, cubic, and quartic the CoV is detectable at $t=0$ and firing immediately is optimal; a one-line rule matches or beats the policy on all three. We say so explicitly.
>
> (ii) On the exponential family, the natural rule scores zero. This is the only family requiring a nested CoV. The naive rule solves $0/75$: the failure is not detection but timing — the rule applies the substitution to an unexpanded form and the episode ends in a rational form the closed actions cannot finish. The learned policy expands first, then fires, solving $56/75$ from reward alone. A rule matches it ($57/75$) only once given that precondition, which we obtained by reading the policy's traces.
>
> What we now claim. Contribution (a) is restated from superiority to recovery: the policy recovers, from reward alone, a trigger as good as a hand-engineered detector, without being given the detector. Where a closed-form depression detector exists, a rule suffices; the learned trigger's value is that it requires no such detector — the property that carries to settings (e.g. ODE reductions) where no per-family detector is available.
>
> Trigger accuracy (5 converged seeds, $n=91$): precision $0.90$, recall $0.97$, $F_1$ $0.93$. Recall is near-ceiling; the false-positive rate on closed equations is partly benign ($71%$ of those firings occur on episodes the policy goes on to solve).
>
> RC2 (Would strengthen) — Repeat the ablation with multiple seeds
> Run — and it corrected one of our own rows. Table 11 is replaced by a seed-matched ablation: all four configurations on the same three seeds, same 3M-step budget, fixed decoder, no checkpoint selection. The CoV machinery (cube-root + nested CoV) carries essentially the whole effect ($+0.594$, positive on every seed). The action-diversity penalty is harmful ($-0.245$, negative on every seed; v5 reported $+0.03$, a figure that compared a training peak against a final value across a seed change). The freshness buffer is unresolvable (sign flips).
>
> Removing the penalty ($\alpha=0$, same seeds, same budget) raises the full-stack control from $0.490$ to $0.891$ with $0/9$ stalls; end-to-end lifts $+0.521$/$+0.537$, both positive on every seed. We retain $0.79$ as the headline because all downstream analyses used the $\alpha=0.1$ cohort; $0.79$ is not called a lower bound. The trigger control re-run at $\alpha=0$ leaves the "recovery, not superiority" conclusion unchanged ($58/75$ vs $56/75$ on the exponential family).

---

### Review · Reviewer_rknj · 2026-07-13

**Summary Of Contributions:**

This paper studies reinforcement learning for step-by-step symbolic equation solving. It formulates algebraic manipulation as an MDP with a dynamic, per-state action space, introduces a lightweight tree-structured policy/value network (TreeMLP), and combines it with curriculum sampling, success replay, constant relabeling, macroactions, and value-guided beam search. The paper first evaluates the system on closed symbolic-equation benchmarks, including the CommonCore equations benchmark from ConPoLe, where the reported greedy accuracy is comparable to the prior result. It then considers a restricted class of "open" equations where a change of variables is required, exposing CoV as a macroaction whose substitution is supplied by a supervised generator or by SymPy while the RL policy learns when to trigger it and how to compose it with ordinary algebraic actions. The paper also reports a seed-level bimodality on a larger open-equation benchmark and explores learning-progress curricula as a partial mitigation.

The main strengths are:

- The problem is well motivated: learning verifiable algebraic transformation traces is meaningfully different from one-shot symbolic solvers and from using a CAS only for the final answer.
- The MDP formulation and dynamic action-space design are concrete and useful for readers interested in procedural symbolic reasoning.
- The paper is unusually transparent about several limitations, including the restricted nature of the open-equation families, the reliance on decode-time beam search, small-seed uncertainty, and the limited statistical support for the UCB-LP mitigation claim.
- The decomposition of the open-equation pipeline is helpful, especially the comparison between the learned CoV generator and SymPy substitution.
- The paper includes detailed appendices with hyperparameters, solution traces, ablations, per-class breakdowns, and compute information.

The main weaknesses are:

- Several headline claims are stronger than the evidence supports. In particular, "from reward alone" and "open equation solving" can be misleading unless consistently tied to the restricted benchmark and to the fact that the CoV substitution generator is supervised or replaceable by a CAS.
- The strongest open-equation result depends heavily on value-guided beam search at evaluation time. The greedy policy remains much weaker, so the paper should more clearly separate learned policy quality from decode-time search.
- The evidence for several design choices and interventions is not always statistically controlled. Some ablations are single-seed, Table 2 is not like-for-like across rows, and the UCB-LP escape-rate improvement is explicitly not significant at the reported sample sizes.
- The open benchmark is narrow: polynomial families are restricted to depressable forms, the CoV generator is evaluated mainly on symbol permutations, and generalization to genuinely new substitution structures is untested.

**Audience:**

Yes

**Audience Explanation:**

Yes. The paper should interest readers working on reinforcement learning for symbolic reasoning, procedural reasoning, program-like action spaces, and neural-symbolic systems. Even though the domain is CAS-solvable, the focus on generating verifiable transformation traces is relevant to settings where the process matters, such as tutoring systems, theorem-prover tactics, and structured mathematical workflows.

The paper also contributes empirical observations that are useful beyond this exact equation-solving setup: the large gap between greedy policy performance and beam-decoded performance, the seed-level bimodality on the larger benchmark, and the class-level policy-trap analysis are all relevant to sparse-reward RL in structured symbolic environments. The work is not yet a broadly general open-equation solver, but it is a useful controlled case study in how far an RL policy plus search can go in a symbolic manipulation environment.

**Broader Impact Concerns:**

None. The work studies symbolic equation solving in a controlled mathematical environment. I do not see specific ethical or societal risks that would require a broader-impact revision beyond the short statement already included. The main broader-impact issue is positive and methodological: if extended, verifiable step-by-step symbolic solvers could be useful in education and formal reasoning tools.

**Claims And Evidence:**

Yes

**Claims Explanation:**

The central evidence is mostly accurate and clearly reported, but several claims need more careful wording and some additional analysis before I would regard the submission as fully convincing.

The closed-equation result is reasonably supported: the paper evaluates on the CommonCore split used by ConPoLe and reports matched greedy decoding, while also giving ablations showing the importance of TreeMLP, constant relabeling, and success replay. The open-equation result is also supported in the restricted benchmark considered by the paper: the authors report held-out performance, per-class behavior, value-guided beam results, a comparison against uninformed or heuristic search, and a decomposition where replacing the learned CoV substitution with SymPy has little end-task effect.

However, the strongest claims require qualification. First, the phrase "from reward alone" is only true for the main policy's trigger/composition behavior, not for the full open-equation system, because the CoV substitution generator is supervised and trained on known target substitutions. The paper often acknowledges this, but the abstract, introduction, and conclusion should be more consistently precise. Second, the "open" setting is restricted to families where a known CoV exists and the benchmark excludes generic polynomial cases; this is an acceptable experimental scope, but the headline language should keep the restricted-open qualifier close to the claim. Third, the reported open-equation success is mainly a beam-search result: the paper reports open_small beam accuracy of 0.79 but greedy accuracy around 0.24. This supports the claim that the learned policy informs search, but it is weaker evidence that the policy alone has learned robust step-by-step solving. Fourth, some ablations and intervention claims are underpowered. The paper is admirably explicit that several comparisons are single-seed diagnostics and that the UCB-LP escape-rate improvement is not statistically significant, but the main narrative should not lean too strongly on these trends.

Overall, I would answer "Yes" because the paper generally provides evidence for the appropriately qualified version of its claims, and it is transparent about many caveats. The revision should make those qualifications part of the main claim, not only part of the limitations and table captions.

**Requested Changes:**

Critical to acceptance:

- Qualify the main "reward alone" claim throughout the paper. The current system for open equations includes a supervised CoV substitution generator, and the authors also show that SymPy can replace that generator with little end-task effect. The learned contribution is the trigger and composition policy, not the generation of the substitution itself. This distinction is stated in places, but it should be made explicit wherever the paper claims reward-only learning for open equation solving.
- Keep the "restricted open" scope close to all headline open-equation claims. The benchmark uses controlled families where a suitable change of variables exists, and it excludes generic polynomial cases. This scope is scientifically reasonable, but the abstract, introduction, contribution list, and conclusion should avoid language that could be read as solving open symbolic equations in general.
- Separate policy learning from decode-time search more sharply. The reported open_small result is 0.79 with beam search while greedy performance is around 0.24. The paper should make this dependency prominent in every headline result table or claim, report greedy and beam numbers side by side where possible, and discuss what errors remain in the greedy policy. This is important because the contribution seems to be policy-guided search rather than a reliably greedy solver.
- Strengthen or further qualify the statistical evidence for ablations and interventions. Some ablation rows are single-seed diagnostics, Table 2 compares single-seed rows against a 5-seed mean, and UCB-LP is selected after an initial screen and has a non-significant escape-rate change. The paper is transparent about these issues, but the main narrative should avoid implying component-level effect sizes or a demonstrated mitigation. If feasible, add a small number of matched-seed ablations for the most important components, especially value-guided beam, freshness-managed replay, and UCB-LP.
- Clarify checkpoint selection and test-set usage. The open_small held-out slice contains only 64 equations after the 30/70 split, while validation-based checkpoint selection is performed across saved checkpoints. Please report final-checkpoint performance alongside validation-selected performance, or otherwise show that checkpoint selection does not materially inflate the reported results on both open_small and open_large.
- Expand the discussion of πcov generalization. The reported 100% result is on held-out symbol permutations within known template families, not on new substitution structures. This is fine as an engineering component, but the paper should explicitly avoid presenting πcov as evidence of general CoV generation. A useful addition would be an experiment or qualitative failure analysis on held-out coefficient-symbol pools, perturbed templates, or closely related CoV forms outside the training grammar.

Would strengthen the work:

- Add confidence intervals or per-seed distributions for the main closed-equation comparisons, not only for the open_large intervention sweep. This would help readers judge whether the CommonCore and small/large closed-equation results are stable.
- Provide a clearer accounting of search cost versus success rate. The paper mentions that policy-guided beam is about 7x cheaper than BFS in one comparison, but a compact table with expansions, wall-clock cost, and accuracy for greedy, beam widths, BFS, and heuristic search would make the learned-search contribution easier to evaluate.
- Include more failure examples from the weak abel_level3 class. Since this class remains weak even for escape seeds, a few representative failed traces would clarify whether the remaining issue is reward shaping, action-set design, value estimation, or search.
- Clarify the relationship between the custom closed-equation benchmarks and the open benchmarks. The paper uses several datasets with similar names and related but different distributions; a single dataset summary table with sizes, train/test splits, equation families, and whether each is closed, restricted-open, or mixed would improve readability.
- Tone down claims about ruling out alternative explanations for bimodality. The probes against dormant neurons, data skew, and encoder collapse are useful, but they are not exhaustive. The current "ruled out" language should be replaced with wording such as "we found no evidence for" or "these probes make the following explanations less likely."
- Release code and datasets before or during the review discussion if possible. The environment is complex enough that reproducibility depends on exact action masks, SymPy simplification behavior, checkpoint selection, beam-search implementation, and crash handling.

---

> ### Author Response · Authors · 2026-07-20
> **Reply to reviewer rknj**
>
> We thank the reviewer for a careful and constructive report, and for finding the core claims supported and the problem of interest. We agree with the requested changes; most are already addressed in the current manuscript (v7), and the two remaining gaps are fixed in v8 (the revision accompanying this response). We address each point below.
>
> Critical requests
> RC1 — Qualify "reward alone" throughout.
> Addressed in v7. The abstract, introduction, contributions, and conclusion all now read: "the main policy trained from reward alone (the CoV substitution generator is supervised but interchangeable with a CAS call)." The distinction between the main RL policy and the supervised generator is stated at every headline claim site.
>
> RC2 — Keep "restricted open" close to all headline claims.
> Addressed in v7. "Restricted-open" appears consistently in the abstract, introduction, contributions, and conclusion. The benchmark scope (hand-designed families with known CoV forms, generic polynomials excluded) is stated in Section 4.2 ("Benchmark scope" paragraph).
>
> RC3 — Separate policy learning from decode-time search more sharply; greedy ≈ 0.24.
> We need to flag an important correction here. The reviewer cites greedy ≈ 0.24 — this was a stale number from an earlier cohort (pre-$\pi_{\mathrm{cov}}$ fix) that was corrected in v5. The true 5-seed greedy mean is $0.67$ (range $0.40$--$0.80$), reported in the abstract, Section 4.2.2, and Table 2 of v7. This inverts the concern: the greedy top-1 policy alone ($0.67$) exceeds every non-learned search baseline (BFS $0.26$, A$^*$ $0.38$--$0.64$), so the result is policy-dominated, with value-guided beam adding only $\approx 0.12$. Table 2 makes the three-way greedy/beam/search comparison explicit, and the beam-width ablation (Figure 2) shows accuracy rising from greedy through width 5 and then saturating. We apologise that the stale number was visible in an earlier version; the corrected figure has been in the manuscript since v5.
>
> RC4 — Strengthen statistical evidence; seed-matched ablations.
> Addressed in v7. The single-seed cumulative ablation table is replaced by a seed-matched ablation (4 configurations × 3 shared seeds × 3M steps, fixed decoder, no checkpoint selection; Table 11). Paired within-seed increments are reported. The CoV machinery carries the whole effect ($+0.594$, positive on every seed); the action-diversity penalty is harmful ($-0.245$, negative on every seed); the freshness buffer is unresolvable (sign flips). UCB-LP is described uniformly as a non-significant positive trend (Fisher exact $p \approx 1.0$), not a demonstrated fix.
>
> RC5 — Clarify checkpoint selection and test-set usage.
> Addressed in v7. The "Unless stated otherwise" sentence at the top of Section 4.2.2 states the 30/70 val/test split and validation-based checkpoint selection protocol; the full protocol is in Appendix A.14. The seed-matched ablation (Table 11) uses fixed final-step evaluation with no checkpoint selection, and is labelled as such. On \texttt{open_small} the val-best result is within $\sim 0.02$ of the final-step value (monotone-to-plateau learning), which we note in the appendix protocol section.

---

### Author Response · Authors · 2026-07-08
**General response (to all reviewers)**

We thank both reviewers for careful, constructive reports, and for finding the core claims supported (both "Yes / Yes") and the problem of interest to the TMLR audience. We agree with essentially every requested change. The unifying theme across both reviews — that some high-level wording read more broadly than the evidence warrants — is real, and the revision narrows the language to match the empirical scope while adding the missing formal semantics and a sharper policy-vs-search analysis. Three cross-cutting changes are worth stating once, up front:

1. **Claims narrowed, priority corrected.** Every "first" claim now attaches to the **restricted-open (change-of-variables)** setting, not to nonlinear solving. We credit O'Keeffe (2025, arXiv:2510.17022) with prior RL for nonlinear *closed* equations and add a dedicated comparison paragraph (Section 2). "From reward alone" is scoped to the main policy (the CoV substitution generator π_cov is supervised), "open" is consistently "restricted-open," and "solving" is the one-verified-root criterion.

2. **A stale headline number corrected — and it strengthens the paper.** Both reviewers flagged the greedy/beam gap (greedy reported as ≈0.24). In building the requested three-way comparison we found the greedy figure was **stale** — inherited from a pre-π_cov-fix cohort; the beam number had been refreshed to 0.79 but greedy had not. The **true 5-seed greedy is 0.67** (range 0.40–0.80), re-measured on the same cohort that yields beam 0.79 and confirmed on additional converged checkpoints (greedy 0.76–0.82). This inverts the concern: the greedy **top-1 policy alone (0.67) exceeds every non-learned search** (BFS 0.26, A* ≤0.64), so the accuracy is **policy-dominated**, with value-guided beam adding a modest ≈0.12. New **Table 2** makes this explicit; the corrected number now appears in the abstract, introduction, Section 4.2.2, and conclusion.

3. **UCB-LP downgraded.** The learning-progress curriculum is now consistently described as "a candidate mitigation direction showing a non-significant positive trend" (Fisher exact p ≈ 1.0), never as a demonstrated fix.

The per-reviewer responses below address each requested change individually.